

**TITLE**

Comparing Stability in Random Forest Models to Map Northern Great Plains Plant Communities
in Pastures Occupied by Prairie Dogs Using Pleiades Imagery

Jameson Brennan[a,], Patricia Johnson[a], and Niall Hanan[b]

[a] South Dakota State University West River Agricultural Center 1905 N Plaza Dr. Rapid City,

SD 57702

[b] Jornada Basin LTER, New Mexico State University Plant and Environmental Sciences Las

Cruces, NM 88003

Corresponding author: Jameson Brennan

Email: Jameson.brennan@sdstate.edu

Second Author email: Patricia.johnson@sdstate.edu

Third Author email: nhanan@ad.nmsu.edu

**ABSTRACT**

Black tailed prairie dogs (*Cynomys ludovicianus)* have been described as a keystone species and important for grassland conservation, yet many concerns exist over the impact of prairie dogs on plant biomass production and consequently livestock production.  The ability to map plant communities in pastures colonized by prairie dogs can provide land managers with an opportunity to optimize rangeland production while balancing conservation goals.  The aim of this study was to test the ability of random forest (RF) to classify five plant communities located on and off prairie dog towns in mixed grass prairie landscapes of north central South Dakota, assess the stability of RF models among different years, and determine the utility of utilizing remote sensing techniques to identity prairie dog colony extent.  During 2015 and 2016, Pleiades satellites were tasked to image the study site for a total of five monthly collections each summer (June-October).  Training polygons were mapped in 2016 for the five plant communities and used to train RF models.  Both the 2015 and 2016 RF models had low (1%) out of bag error rates.  However, comparisons between the predicted plant community maps using the 2015 imagery and one created with the 2016 imagery indicate over 32.9% of pixels changed plant community class between 2015 and 2016.  The results show that while RF models may predict with a high degree of accuracy, overlap of plant communities and inter-annual differences in rainfall may cause instability in fitted models.  A final RF model combining both 2015 and 2016 data yielded the lowest error rates, and was also highly accurate in determining prairie dog colony boundaries.

**Keywords**

Remote sensing, random forest, rangelands, plant ecology, high resolution imagery

**INTRODUCTION**

Within the Northern Great Plains mixed grass prairie ecosystem, black tailed prairie dog colonization is an issue of concern for livestock producers (Miller et al. 2007). Competition between prairie dogs and livestock is a major concern for land managers looking to optimize beef production while still conserving wildlife species (Augustine and Springer 2013). Prairie dogs have been identified as a keystone species, and are often seen as ecosystem engineers providing habitat to a number of other plant and wildlife species (Davidson et al. 2010; Kotliar et al. 1999). Prairie dogs can also reduce availability of forage for livestock by directly reducing the quantity of forage available (through direct consumption, clipping plants to increase predator detection, and building soil mounds), and by changing species composition (Derner et al. 2006). Within the mixed grass prairie, C3 mid-grasses tend to decrease and C4 short-grasses increase along an increasing gradient of grazing intensity (Irisarri et al. 2016). Due to repeated defoliation, older core areas of prairie dog towns often become characterized by extensive areas of bare ground and low vegetation production, which is generally limited to annual forb and dwarf shrub species. Pastures containing extensive areas of bare ground due to prairie dog colonization may potentially depress livestock forage intake rates and ultimately beef production. The ability to map the extent and monitor the impact of prairie dogs on the landscape can help land managers looking to optimize livestock production on prairie dog occupied rangelands.

Remote sensing of rangelands greatly improves our ability to study and understand complex ecological interactions across the landscape. As technology advances, monitoring of rangelands via remote sensing platforms will facilitate research products freely available to land managers (Browning et al. 2015). One of the main advantages of remote sensing data is its capacity to cover wide areas, allowing assessment of plant communities at landscape level scales

as compared to traditional point-based assessments (Ramoelo et al. 2015; Yu et al. 2018).
Numerous studies have demonstrated the utility of remote sensing applications in monitoring
rangeland condition, including mapping of vegetation communities, plant species composition,
biomass estimation, and impact of grazing intensity on the landscape (Goodin and Henebry
1997; Blanco et al. 2008; Franke et al. 2012).

Many methods for accurately classifying plant communities using remote sensing

techniques have been used in ecological and natural resource studies. One method, random forest
classification (RF), has gained considerable traction in the remote sensing community for its
ability to produce accurate classifications, handle highly dimensional data, and provide efficient
computing times (Belgiu and Drăguţ 2016).  RF is seen as an improvement over simple
classification tree analysis by reducing noise and misclassification of outliers (Laliberte et al.
2007; Nitze et al. 2015).  RF is an ensemble decision tree classifier which combines bootstrap
sampling to construct several individual decision trees from which a class probability is assigned
(Mellor et al. 2013).  RF builds each tree using a deterministic algorithm selecting a random set
of variables and a random sample from the calibration data set (Ramoelo et al. 2015).

The utility of random forest algorithms has been demonstrated in remote sensing

applications across many plant communities at multiple scales (Mutanga et al. 2012; Lowe and
Kulkarni 2015; Ramoelo et al. 2015).  Concerns exist, however, over the transferability of these
models to different sites, across seasons, or years.  For example, RF models have shown to have
a high degree of classification accuracy for mapping fine scale coastal vegetation using digital
elevation maps and high resolution orthophoto imagery, but model accuracy decreased
significantly when applied to spatially separated sites (Juel et al. 2015).  Selecting spatially
releveant training data or including species level cover data may help improve or explain
differences observed when transferring models between sites.  Incorporating additional seasons
of data may also improve RF model accuracy; previous research has shown an improvement of
RF model accuracy in classifying wetlands in northern Minnesota with the inclusion Landsat 5
images across two years using full season data versus summer only, and fall only models
(Corcoran et al. 2013).  Longer term studies have also demonstrated the utility of using RF
modeling with 30m Landsat data to monitor rangeland cover across the western United States
over a 33 year period (Jones et al. 2018).  Results of these studies suggest the scale and
seasonality of the imagery may play an important role in the stability and accuracy of RF
models.

The stability in RF models to accurately map plant communities within prairie dog

occupied pastures may be particularly important for managers looking to monitor prairie dog
colony expansion or contraction over time.  While classification rates are often reported in
studies, the potential overlap in plant community composition is rarely explored as a potential
source of error within the models.  Many research studies focus solely on spectral differences in
plant communities and fail to analyze community differences on the ground at the species level
(de Colstoun et al. 2003; Geerken et al. 2005).  This may be especially important within prairie
dog occupied rangelands, where shifts in plant community composition may be driven more by
the presence or absence of an herbivore species versus elevation, soils, or other landscape
features.  These herbivory induced changes in plant community may facilitate or hamper
classification schemes.  The ability to accurately map plant communities within prairie dog
occupied pastures can help improve management of rangelands colonized by prairie dogs, yet
little research has explored the possibility of utilizing remote sensing as a tool to do so.
A large collaborative study from 2012-2016 was conducted to evaluate livestock
production on mixed-grass prairie pastures with varying levels of prairie dog occupation.  A
major goal of the larger study was to determine which plant communities on the pastures cattle
preferred to graze, and how those preferences shifted within and between years (Olson et al.
2016). Plant communities on the site were categorized based on location (on- or off-town) and
visually apparent dominant plant functional groups.  Thus, plant community as defined for this
study was a collection of species within an area of a relatively uniform composition different
from neighboring patches.  Differences in neighboring patches were evident by differences in
dominant functional group (forb vs grass) or differences in photosynthetic pathways (C3 vs C4
grasses).  The overall goal of this paper, then, was to develop maps that accurately classify plant
communities based on satellite imagery collected between years. Specific objectives of this study
were to 1) determine differences in the five identified plant communities based on species
composition, 2) assess the utility of using a RF model with high resolution satellite imagery to
classify plant communities of interest within a mixed grass prairie ecosystem containing prairie
dogs, 3) determine the stability of the RF model when using subsequent years of satellite
imagery with identical training data, and 4) determine the ability of high resolution satellite
imagery to accurately map prairie dog towns. Our ability to map and understand these plant
communities' at large scales will give researchers insight into applying RF models across years
using high resolution imagery.  Research from this study will allow us to better assess how
prairie dogs drive changes in plant communities, and provide a new tool to map the extent and
impact of prairie dog colonization on the landscape to better inform land management decisions.

**METHODS**

**Study site**
The study area (45.74N, 100.65W) was located near McLaughlin, South Dakota on a
northern mixed-grass prairie ecosystem.  Native prairie pastures (810 ha total area) were leased
from 2012-2016; pastures were continuously stocked with yearling steers from June-October of
each year to achieve 50% utilization.  Of the 810 ha, approximately 186 ha were occupied by
black-tailed prairie dogs (*Cynomys ludovicianus*).  Predominant soils at the site were clays and
loams. Ecological sites, and the plant communities they support vary widely; Loamy and Clayey
were the predominant Ecological Sites at the site with inclusions of Dense Clay, Shallow Clay,
and Thin Claypan (Barth et al. 2014).  Plant species dominating the site were largely native,
including western wheatgrass (*Pascopyrum smithii* Rydb.), green needlegrass (*Nassella viridula*
Trin.), and needle-and-thread (*Hesperostipa comata* Trin. & Rupr), intermixed with blue grama
(*Bouteloua gracilis* Willd. Ex Kunth), buffalograss (*Bouteloua dactyloides* Nutt.), and sedges
(*Carex* spp.). The most common non-native species on the site was Kentucky bluegrass (*Poa*
*pratensis* Boivin & Love). Woody draws occupied moist drainage areas; vegetation consists
primarily of bur oak (*Quercus macrocarpa* Nutt.), American plum (*Prunus americana* Marshall),
and chokecherry (*Prunus virginiana* L.). These draws were frequently flanked by snowberry-
dominated patches (*Symphoricarpos occidentalis* Hook.).  Plant communities on areas occupied
by prairie dog towns on the site were largely dominated by western wheatgrass and shortgrasses
(buffalograss, blue grama, and sedges) intermixed with patches of bare ground and annual forb
dominated areas.  Common annual forbs on prairie dog towns included prostrate knotweed
(*Polygonum aviculare* L.), fetid marigold (*Dyssodia papposa* Vent.), dwarf horseweed (*Conyza*
*ramosissima* Cronquist), and scarlet globemallow (*Sphaeralcea coccinea* Nutt.).  A weather
station has been maintained on site from May 2013 operated by South Dakota Mesonet.  Mean
annual rainfall at the site is 446 mm and average growing season (May through September)
temperature is 15.3ºC (South Dakota Climate and Weather 2017).
Five plant communities of interest for our study site were identified: 1) Forb-dominated
sites on prairie dog towns (On-Forb), 2) Grass-dominated sites on prairie dog towns (On-Grass),
3) Snowberry-dominated sites off-town (Off-Snow), 4) Cool season grass-dominated sites off-
town (Off-Cool), and 5) Warm season-dominated sites off-town (Off-Warm).  An additional
plant community labeled 'Draws' was delineated visually within ArcGIS software due to
difficulty in mapping these areas in the field.  Areas delineated as Draws were removed from the
analysis area.
**Training sites**
To facilitate classification, training site polygons were mapped for On-Forb, On-Grass,
Off-Cool, Off-Warm, and Off-Snow plant communities using ArcPad for Trimble GPS units in
the summer of 2016.  Twenty training sites were mapped for each of the plant communities
except Off-Warm, for which only 8 sites were mapped due to the difficulty of finding
homogenous stands of warm season grasses. Plant species in the Northern Great Plains are
dominated by cool season species; warm season species, where they occur, are typically
intermixed into stands of cool season species. Training sites for each plant community were
selected from across the entire study area to capture potential site differences across research
pastures.  Sites were mapped in the field by walking the perimeter of the plant community patch
with a Trimble GPS unit.  Training polygon perimeter boundaries were always at least 3 meters
interior of patch edge to minimize error introduced to the training data as a result of GPS signal
noise. Identified patches were then converted into a polygon shapefile within ArcGIS to be used
as training polygons for the RF classification algorithm.  Within each training site polygon, three
0.25 m$^2$ plots were randomly located by tossing plot frames into the area of interest to determine
sampling area.  Within each plot, percent cover by species was recorded in the summer of 2016
at the time of polygon mapping.

**Plant Community Analysis**

Plant community analysis was performed on vegetation data collected from the three

0.25m$^2$ plots measured in each training polygon. Differences between plant community
compositions were determined using a Multi-Response Permutation Procedure (MRPP) with the
Sorensen Bray-Curtis distance method.  MRPP is a nonparametric procedure used for testing
hypotheses between two or more groups (Mitchell et al. 2015).  Differences in community
compositions were analyzed for all plant communities, and pairwise comparisons generated.  To
analyze trends in species composition between plant community plots, Non-metric
Multidimensional Scaling (NMS) ordination was used (Kruskal 1964).  Only species that
occurred in 3 or more plots were included in the ordination analysis.  NMS analysis was
conducted using the Sorensen Bray-Curtis distance method with 250 iterations and a stability
criterion of 0.00001.  Analysis was repeated five times to confirm ordination pattern in the data.
Similarity index matrices were generated to compare plot differences between plant communities
and averaged by plant community.  All ordination analyses (MRPP and NMS) were performed
using PC-ORD 6 software (McCune and Mefford 2002).

**Imagery**

During the summers of 2015 and 2016, Pleiades satellites were tasked to image the study

site.  Pleiades satellites, which are members of the SPOT family of satellites, are operated by
AIRBUS Defense and Space.  This platform was chosen due to its high spatial resolution (0.5 m
pan chromatic, 2 m multispectral) and four band spectral resolution: pan chromatic (480-830
nm), red (600-720nm), green (490-610 nm), blue (430-550 nm), and near infrared (750-950 nm).
Pleiades satellites were designed for commercial tasking and monitoring, allowing multiple
revisits to a project site.  A total of ten image collections were acquired in the summer of 2015
and 2016 (five each year) from June through October during the 1$^{st}$-15$^{th}$ of each month (Table 1).
Image collection times were chosen to correspond to the time periods when cattle were actively
grazing on the site.  Multispectral images were pan-sharpened and orthorectified by the image
provider (Apollo Imaging Corp).  Each monthly image collection was converted into an NDVI
image.  In addition, boundaries of the prairie dog town were mapped using a handheld Trimble
GPS unit to compare predicted colony location with ground truth location.
**Random Forest model**

For the RF model, the Random Forest package of the Comprehensive R Archive Network

(CRAN) implemented by Liaw and Wiener (2002) was utilized.  Training data were constructed
by stacking all satellite imagery spectral bands (Red, Blue, Green, and NIR) and NDVI bands for
each month of each year (25 total dimensions per year) to create a raster stack for each year's
imagery (2015 and 2016).  To train the model, pixel values were extracted from the satellite
imagery raster stack for each training polygon mapped in the field.  The random forest models
were built using 200 decision trees and default number of nodes at each split (sqrt(n)), with plant
community data as the response category (On-Grass, On-Forb, Off-Cool, Off-Warm, and Off-
Snow) and spectral band values as the predictor.  Models were checked for error stabilization, for
all models error rates stabilized around 50 trees.  Yearly models (2015 and 2016) were built for
output comparison.  A combined years model was also constructed using all available spectral
data from 2015 and 2016 (50 dimensions).
Within the random forest package, Out of Bag (OOB) error rates were calculated by
reserving one-third of the training data to test the accuracy of the predictions.  Models were then
used to predict class belonging for 2015 and 2016 raster stacks and the combined 2015 and 2016
stack using the 'predict' function within program R.  To assess the stability of the RF models
from year to year, the "crosstab" function in the raster package in program R was used to
calculate the number of pixels that changed class from 2015 to 2016.  The output was used to
calculate percent of pixels that were unchanged from 2015 to 2016 model predictions and
percent of pixel change that occurred between years for plant community predictions.

**Results and Discussion**

**Plant Community**
MRPP pairwise comparisons results showed a significant difference between all plant
communities ($P < 0.001$).  Differences are evident between plant communities in the 2-D plot of
the NMS ordination (final stress = 20.01, instability $< 0.00001$ after 66 iterations), with some
overlap occurring between communities (Figure 1).  Plant communities on-town and off-town
are clustered at opposite ends of the ordination plot, with the greatest distance being between On-
Forb and Off-Snow.  Detrended correspondence analysis of plant communities ranging from
uncolonized, 2 years post colonization, and 4-6 years post colonization showed that uncolonized
sites were clustered at one extreme and the 4-6 year sites at the other extreme (Archer et al.
1987).  Interestingly, Off-Warm and On-Grass communities are clustered closer in ordination
space.  Plant communities shifts on-town towards those dominated by shortgrass species have
been documented (Agnew et al. 1986; Koford 1958), and is probably attributable to the high
grazing resistance of the C4 species blue grama and buffalograss (Derner et al. 2006).

Similarity index differences between plant communities were generated from a Sorensen

(Bray-Curtis) distance matrix, and can be seen in Table 2.  While there is some overlap between
plant communities, in general similarities are low ($< 29\%$), with the greatest distance occurring
between the On-Forb communities and the off-town communities (Table 2).  Based on how plant
communities were selected in this study, we expected plant community composition to be
distinct between groups.  Though plant communities are defined by dominant functional group in
this study, the amount of overlap occurring demonstrates that other functional groups and species
exist within these distinct patches, which may be a potential source of instability in classification
models.
**Random Forest Model Results**

Results from the RF models show low OOB misclassification error rates for each

individual plant community (Table 3) indicating a high degree of accuracy in the model.  Overall
the OOB model error rates were 0.9% and 1.12% for the 2015 and 2016 model respectively.
OOB accuracy is an unbiased estimate of the overall classification accuracy eliminating the need
for cross-validation (Breiman 2001).  OOB error rates have been shown to be reliable estimates
of class accuracy for identifying invasive species (Lawrence et al. 2006), and mapping corn and
soybean fields across multiple years (Zhong et al. 2014).  Belgiu and Drăguţ (2016) in their
review of RF applications in remote sensing acknowledge that the reliability of OOB error
measurements needs to be further tested using a variety of datasets in different scenarios
Consistency in error rates for plant communities appears to indicate stability in the 2015 and
2016 RF models which used identical training sites on consecutive yearly satellite imagery.
However, when comparing yearly predicted plant community maps, differences between
community classifications are slightly more pronounced, indicating the models may not be as
stable as predicted based solely on the OOB error rates.

Overall a total of 67.04% pixels remained unchanged in their plant community

classification from 2015 to 2016 (Table 4).  Of the pixels that changed classification between
years, 15.13 were on-town to off-town transitions, 2.26 were on-town to on-town transitions, and
15.57 were off-town to off-town plant community transitions.  It is unlikely in this northern
mixed-grass prairie ecosystem that all the changes in plant communities indicated by
classification of pixels were real changes from one plant community type to another over one
year.  In the absence of a major disturbance event, such major shifts in species composition
typically occur much more slowly (Vermeire et al. 2018). The results from the plant community
analysis indicate training sites were chosen appropriately to account for differences in species
composition on the ground, therefore apparent changes are much more likely due to factors that
affect the spectral signature of the vegetation.  Factors that may potentially affect spectral
signatures could include changes resulting from prairie dog herbivory, changes in precipitation
regimes, or changes occurring along plant community transition zones.

The pixels changing from On-Grass to Off-Cool represented the highest percentage of

pixels that changed plant community classification at 7.28%; these are likely occurring along
transition zones at the prairie dog colony edge.  Both On-Grass and Off-Cool plant communities
have western wheatgrass as a dominant species.  Similarity in species dominance may explain
some of the challenges to distinguishing between some on and off colony plant communities.
Difficulty in classifying Off-Cool and On-PDG may also be due to subtle vegetation changes
likely induced by the level of herbivory.  Research on a South Dakota mixed grass prairie
showed that prairie dogs remove over four times more biomass than cattle grazing on-town
(Gabrielson 2009).  Up to 7 times more standing dead forage and 60% less standing crop
biomass has been reported on uncolonized sites compared to colonized areas, mainly attributed
to prairie dogs clipping vegetation which greatly reduced the amount of grasses that reached
maturity (Johnson-Nistler et al. 2004).  Areas either less maintained on-town by prairie dogs or
grazed by cattle repeatable off-town may show similar spectral signatures.

Differences in year to year classification could also be attributed to the interannual

variability of rainfall between 2015 and 2016 (Figure 2).  Yearly rainfall patterns can result in
large differences in NDVI and biomass measurements across years (Wehlage et al. 2016).  While
overall total rainfall between years was similar, differences in timing of precipitation that
occurred likely affected timing of green up and dormancy for many of the cool and warm season
species on the site. This, then, would create different NDVI patterns between years (Figure 3).
Goward and Prince (1995) suggested that the relationship between NDVI and annual rainfall in
any given year also depends on the previous year history of rainfall at the site.  Previous research
has shown that annual above ground primary production of shortgrass communities is related to
current as well as previous two years precipitation (Oesterheld et al. 2001).  The above average
rainfall at the study site in 2015 could have added to the increase in average NDVI in 2016 when
compared to 2015 through an increase in cumulative biomass or production at the site.  Increased
cumulative biomass in 2016 may cause higher NDVI values for example in On-PDG plant
communities resulting in classification shifts to Off-Cool; similarly, greater NDVI values in Off-
cool in 2016 may result in some of those pixels being classified as Off-Snow.

Another possible cause for changes in plant community classifications between years is

overlap of species where two communities share a boundary.  One issue with using categorically
classified vegetation maps is that plant communities in space are rarely mutually exclusive, and
tend to change along a continuum with environmental gradients (Equihua 1990).  Plant
communities in the region, which are predominantly comprised of cool season grasses, often
include varying levels of warm season species; and snowberry thickets often have an understory
of grasses, especially near the perimeter.  The challenge of accurately classifying plant
communities along an ecological continuum may be further exacerbated by changes induced by
prairie dogs, where transition zones are less defined by environmental gradients and more
defined by the level of herbivory.  Thus, within and between on-town and off-town plant
communities, transition zones are likely to account for a portion of the classification change
between plant communities between years (Figure 4).  Alternative approaches to mapping plant
communities can be the recognition of fuzzy properties enabling a single point in space to exhibit
characteristics of a number of plant communities (Duff et al. 2014; Fisher 2010).  While fuzzy
classification maps are more likely to give a better picture of plant community composition on a
per pixel basis, they are also more difficult to use to draw inferences of species dominance,
livestock use patterns, and extent of prairie dog colonization.
A final RF model combining all available bands and NDVI values for 2015 and 2016
reduced error rates for all plant communities below 1% (Table 3).  While we have shown that
lower error rates may not result in more stable predictions, using all available data for a model
will likely improve accuracy and result in a more accurate thematic map.  Other studies have
reported increases in classification accuracy in RF models with the addition of combined
seasonal images, hyperspectral data, LiDAR images, radar (SAR) images, and ancillary
geographical data such as elevation and soil types (Corcoran et al. 2013; Pu et al. 2018; Shi et al.
2018; Xia et al. 2018; Yu et al. 2018; Zhou et al. 2018).  RF models have the ability to handle
highly dimensional correlated data, and data combined from multiple different data sources
across different temporal scales; however, one disadvantage to using non-parametric classifiers
such as RF and decision trees is that they require a large number of observations to accurately
estimate the mapping function (James et al. 2014).  Thus the incorporation of additional predictor
variables as well as additional training data will likely result in higher accuracy rates.
The variable importance graph of the combined model indicates that NDVI variables
contribute the most to the model over individual bands (Figure 5).  In classifying vegetation
morphology in a savanna grassland, Mishra and Crews 2014 found spectral classification
features (mean NDVI or ratio NDVI) were the most significant.  The variable importance plot
from the combined data model also indicates that different months between years contribute
highly to the classification accuracy.  Of the ten most important variables in the model, 6 were
from 2015 and 4 from 2016, suggesting additional years' data in the model is likely to yield
greater classification accuracy.  The internal information provided by the model, such as variable
importance, can be a useful tool for researchers to select features of greatest importance to
reduce computation times in the instance of large datasets.  At the size of our study area (810 ha)
and a maximum of 50 variables, the combined 2015-16 data model only slightly added to
computation time, but not enough to warrant feature trimming from the dataset.  Land managers
looking to classify prairie dog colonies on more extensive grasslands may look to including only
NDVI variables into training datasets to increase computational efficiency.
**Remote Sensing Prairie Dog Colonies**
Visual comparison of the predicted on-town plant communities versus off-town plant
communities show a clearly defined boundary between areas colonized by prairie dogs and areas
not colonized (Figure 6).  Results from mapping colony boundaries with a hand held GPS device
estimated the colony to be 276 ha in 2012 to 186 ha in 2015.  Total colony acreage estimated
from summing the pixel area occupied by the On-Grass and On-Forb community pixels from the
combined 2015-2016 RF model was 246 ha.  Previous research has demonstrated that
colonization by prairie dogs and subsequent increases in grazing pressure can result in significant
differences between on- and off-town plant community composition and production (Coppock et
al., 1983; Winter et al. 2002; Johnson-Nistler et al. 2004; Geaumont et al. 2019).  The results of
our study demonstrate that these differences are significant enough to be identified using remote
sensing techniques.  Interestingly, a considerable portion of the area misclassified as on-town is
from a previously colonized area that had been poisoned in 2013, suggesting that, at least
spectrally, these areas still resemble plant communities similar to those actively colonized.  The
higher area estimate from the RF model is likely the result of transition areas controlled two
years prior.  Additionally, most other pixels misclassified as on-town are likely drainage areas
with high bare ground off-town, whose variability was not captured in the dataset.  One prior
study had sought identify prairie dog colonies using 30m Landsat imagery, however concluded
that the scale was too course for accurately measuring prairie dog towns (Wolbrink et al. 2002).
High resolution satellite imagery used in this study appears capable at capturing fine scale
transitions that occur between plant communities along the on-town off-town gradient.

The RF model was also able to accurately predict older core areas of prairie dog towns

(On-forb) often characterized by a high percentage bare ground, low vegetation production, and
dominance by annual forb and dwarf shrub species (Coppock et al., 1983).  Area estimates of
On-Forb were 33 ha and 32 ha in 2015 and 2016 respectively.  State and transition models for
prairie dog towns developed within Custer State Park South Dakota, found older core areas were
considered undesirable for management due to losses of native grasses, increased bare ground,
potential for erosion, extensive presence of exotic species, and increased inputs to restore to a
more desirable state (Hendrix 2018).  The ability to monitor these older core areas of prairie dog
towns remotely may help land managers limit sites from becoming highly degraded, and serve as
a useful tool for land managers concerned over balancing wildlife conservation with losses in
livestock production.
**Conclusions**
Stability of models is important when applying similar techniques across different sites,
plant communities, and in this case years.  Differences in year-to-year NDVI values may alter
classification results, and the addition of two years' worth of data likely resulted in improved
model performance.  One of the main benefits to RF classification in remote sensing is the
relatively fast computing time (Belgiu and Drăguţ 2016), and, given the availability of free
satellite imagery, researchers would be prudent to include multiple images across years and
seasons in their model to improve accuracy.  Furthermore, while the desired outcome is often to
produce thematic maps, recognizing that plant communities rarely exist in discrete communities
is important when selecting community types to map.  Combining plant community ordination
results with remote sensing results can aid in understanding sources of model error and
limitations of classification algorithms.  This is likely to be magnified as pixel size decreases,
resulting in fine scale predictions which may be more susceptible to plant community transitions
zones.  Results from this study indicate that plant community changes induced by prairie dogs
are significant enough to be detected via remote sensing techniques.  Land managers looking to
optimize rangeland health on pastures occupied by prairie dogs may potentially utilize high
resolution imagery to monitor colony size and make recommendations of appropriate stocking
rates based on extent of colonization.

413         **Acknowledgements**

We would like to acknowledge and thank the U.S. Department of Agriculture (Grant
Number 2011-68004-30052) for funding this research as well as North Dakota State University.
We would also like to thank the McLaughlin family for providing access to the land the research
was conducted.

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

**Tables and Figures**

Table 1. Acquisition dates of Pleiades satellite imagery tasked for each month (June – October) in 2015 and 2016.

| 2015 Dates of Acquisition | 2016 Dates of Acquisition |
| --- | --- |
| 6/1/2015 | 6/5/2016 |
| 7/9/2015 | 7/2/2016 |
| 8/4/2015 | 8/2/2016 |
| 9/1/2015 | 9/11/2016 |
| 10/8/2015 | 10/1/2016 |


Table 2. Similarity index (Sorensen (Bray-Curtis) distance method) values averaged by plot
across plant communities.

| Community Comparison[1] | Similarity Index (%) |
|---|---|
| Off-Cool vs. Off-Snow | 28.2 |
| Off-Cool vs. Off-Warm | 27.8 |
| Off-Cool vs. On-PDG | 27.7 |
| Off-Snow vs. Off-Warm | 21.6 |
| On-PDG vs. On-PDF | 17.8 |
| Off-Snow vs. On-PDG | 17.3 |
| Off-Warm vs. On-PDG | 17.3 |
| Off-Cool vs. On-PDF | 7.9 |
| Off-Snow vs. On-PDF | 6.2 |
| Off-Warm vs. On-PDF | 6.2 |

[1]Plant communities on prairie dog towns are grass-dominated (On-Grass) and forb-dominated
(On-Forb); plant communities in off-town areas are cool season grass-dominated (Off-Cool),
warm season grass-dominated (Off-Warm), and snowberry-dominated (Off-Snow).



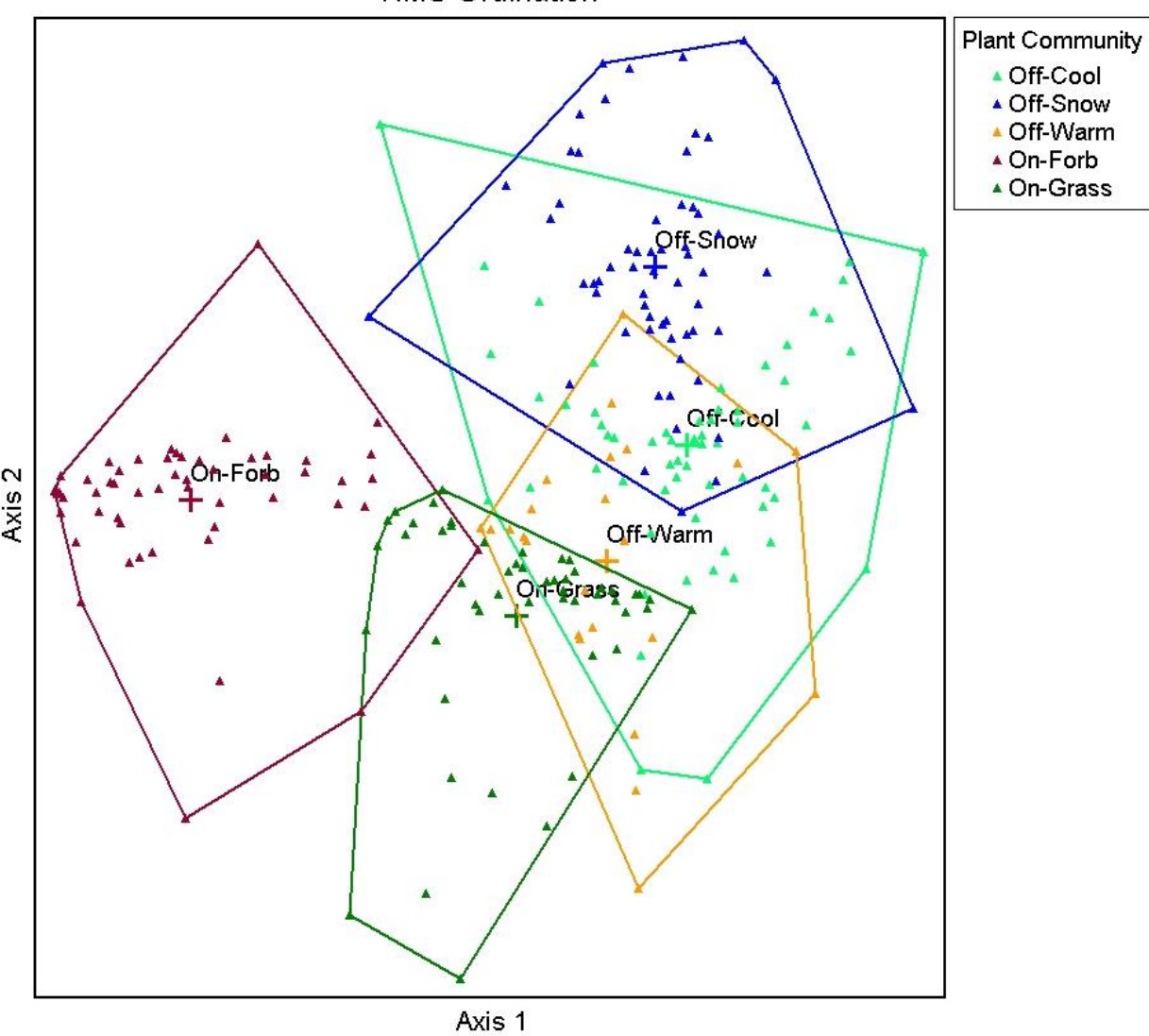


Figure 1. NMS ordination plots for plant communities located on and off of prairie dog towns,
based on plant cover by species data collected in 2016 on the study site in north central South
Dakota.  The '+' symbol followed by the community name represent the weighted mean
(centroid) of the multivariate dataset.  Plant communities on prairie dog towns are grass-
dominated (On-Grass) and forb-dominated (On-Forb); plant communities in off-town areas are
cool season grass-dominated (Off-Cool), warm season grass-dominated (Off-Warm), and
snowberry-dominated (Off-Snow).






Table 3: Out of Bag misclassification error rates (%) for each plant community for 2015, 2016, and combined year random forest models.

| Plant Community[1] | 2015 Model | 2016 Model | 2015-2016 Combined Model |
|---|---|---|---|
| Off-Cool | 0.20% | 0.40% | 0.04% |
| Off-Snow | 2.2% | 1.9% | 0.69% |
| Off-Warm | 3.2% | 5.3% | 0.73% |
| On-Grass | 0.40% | 0.60% | 0.09% |
| On-Forb | 0.60% | 0.70% | 0.19% |

[1] Plant communities on prairie dog towns are grass-dominated (On-Grass) and forb-dominated
(On-Forb); plant communities in off-town areas are cool season grass-dominated (Off-Cool),
warm season grass-dominated (Off-Warm), and snowberry-dominated (Off-Snow).




Table 4: Percent of pixels within each plant community that remain unchanged and that changed

class belonging between 2015 and 2016 models.


| Transition | 2015 PC[1] | 2016 PC | Total Pixels | Percent of Total Pixels |
|---|---|---|---|---|
| | Off-Cool | Off-Cool | 9712857 | 31.03 |
| | On-Grass | On-Grass | 6427817 | 20.54 |
| Unchanged Pixels | Off-Snow | Off-Snow | 3401264 | 10.87 |
| | On-Forb | On-Forb | 887151 | 2.83 |
| | Off-Warm | Off-Warm | 555635 | 1.78 |
| | On-Grass | Off-Cool | 2278390 | 7.28 |
| | Off-Cool | Off-Snow | 1468042 | 4.69 |
| | Off-Cool | On-Grass | 1262373 | 4.03 |
| | Off-Snow | Off-Cool | 1174565 | 3.75 |
| | Off-Warm | Off-Cool | 729511 | 2.33 |
| | Off-Cool | Off-Warm | 716503 | 2.29 |
| | Off-Warm | Off-Snow | 629212 | 2.01 |
| | On-Grass | Off-Snow | 626695 | 2.00 |
| | On-Grass | On-Forb | 362417 | 1.16 |
| Changed Pixels | On-Forb | On-Grass | 343774 | 1.10 |
| | Off-Snow | On-Grass | 281061 | 0.90 |
| | Off-Snow | Off-Warm | 155213 | 0.50 |
| | On-Grass | Off-Warm | 82450 | 0.26 |
| | On-Forb | Off-Cool | 72758 | 0.23 |
| | Off-Cool | On-Forb | 69188 | 0.22 |
| | Off-Warm | On-Grass | 43132 | 0.14 |
| | On-Forb | Off-Snow | 19575 | 0.06 |
| | Off-Warm | On-Forb | 573 | 0.00 |
| | On-Forb | Off-Warm | 314 | 0.00 |
| | Off-Snow | On-Forb | 17 | 0.00 |


[1]Plant communities (PC) on prairie dog towns are grass-dominated (On-Grass) and forb-

dominated (On-Forb); plant communities in off-town areas are cool season grass-dominated

(Off-Cool), warm season grass-dominated (Off-Warm), and snowberry-dominated (Off-Snow).






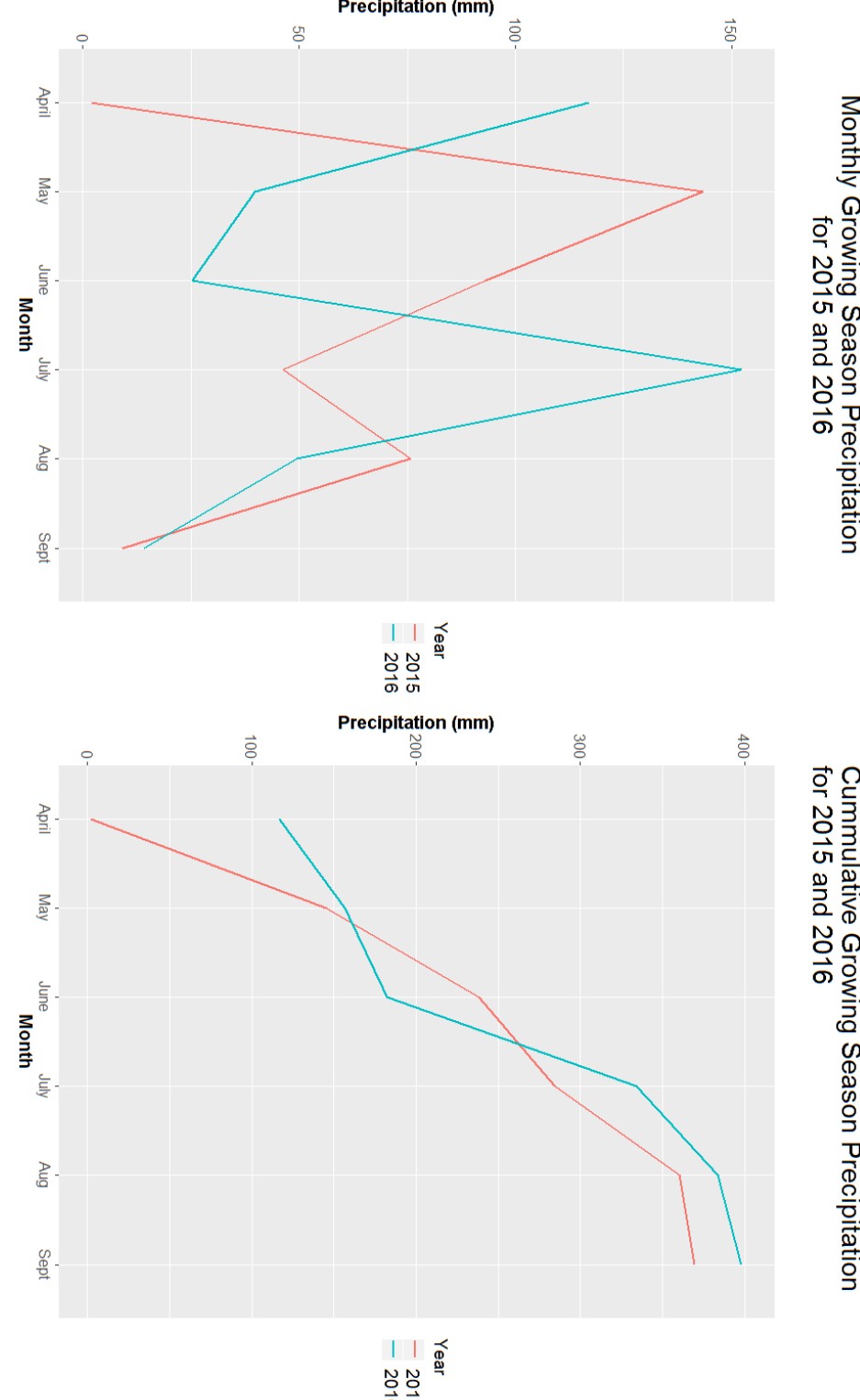

Figure 2: Monthly and cumulative growing season precipitation patterns for 2015 and 2016 recorded at a weather station located on the study area in north central SD (45.737296 N, -100.657540 W)( South Dakota Mesonet 2018).

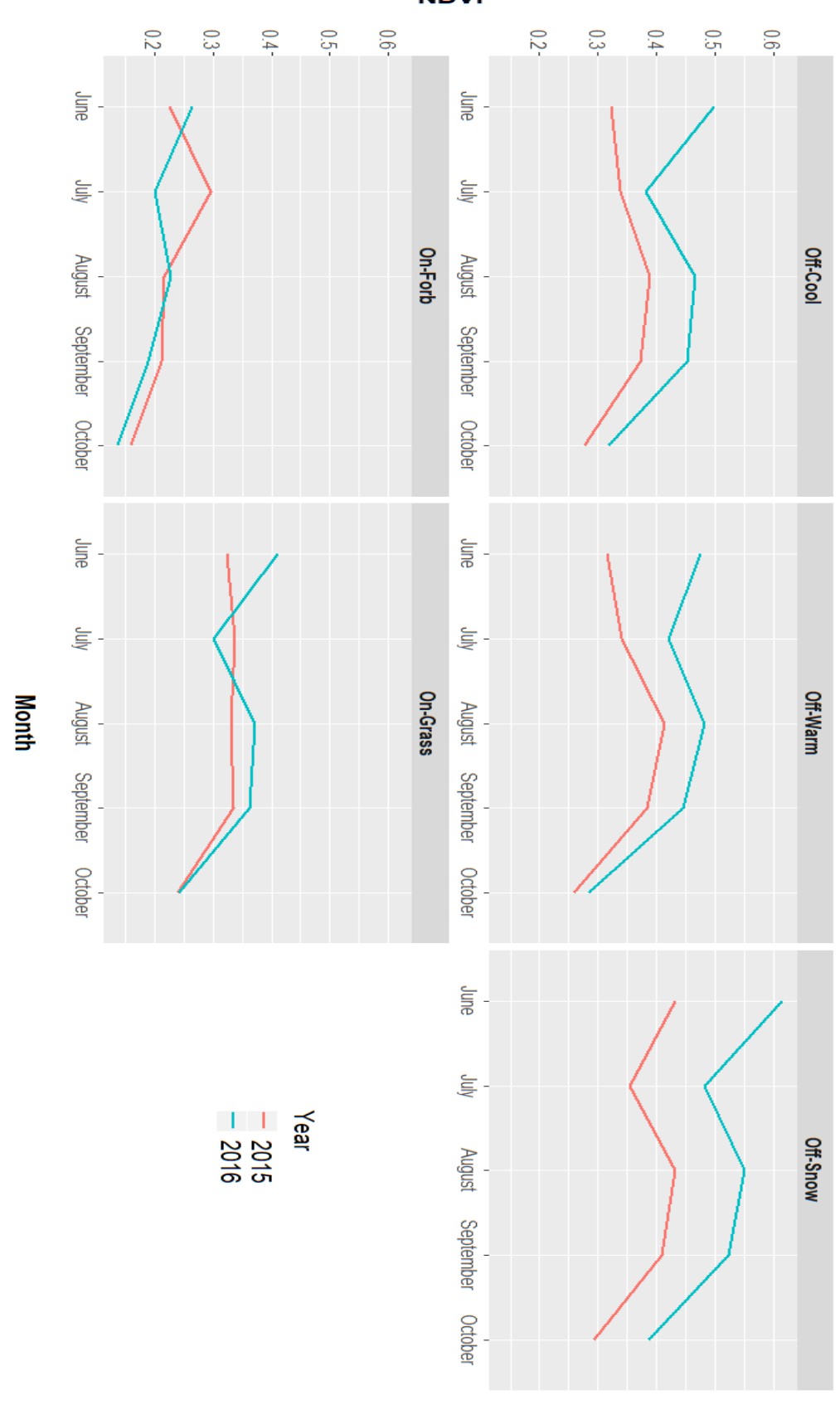

Figure 3: Comparison of mean monthly NDVI for training polygons in five plant communities on the study site in north central SD. Plant communities on prairie dog towns are grass-dominated (On-PDG) and forb-dominated (On-PDF); plant communities in off-town areas are cool season grass-dominated (Off-Cool), warm season grass-dominated (Off-Warm), and snowberry-dominated (Off-Snow).



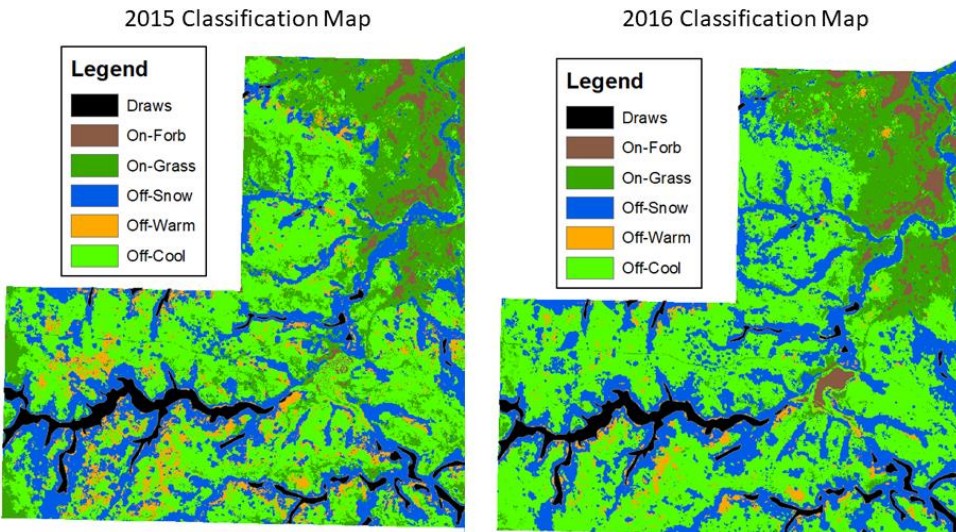


Figure 4: Random forest classification maps from 2015 and 2016 of one pasture in the study area
in north central South Dakota.  Plant communities on prairie dog towns are grass-dominated
(On-Grass) and forb-dominated (On-Forb); plant communities in off-town areas are cool season
grass-dominated (Off-Cool), warm season grass-dominated (Off-Warm), and snowberry-
dominated (Off-Snow).


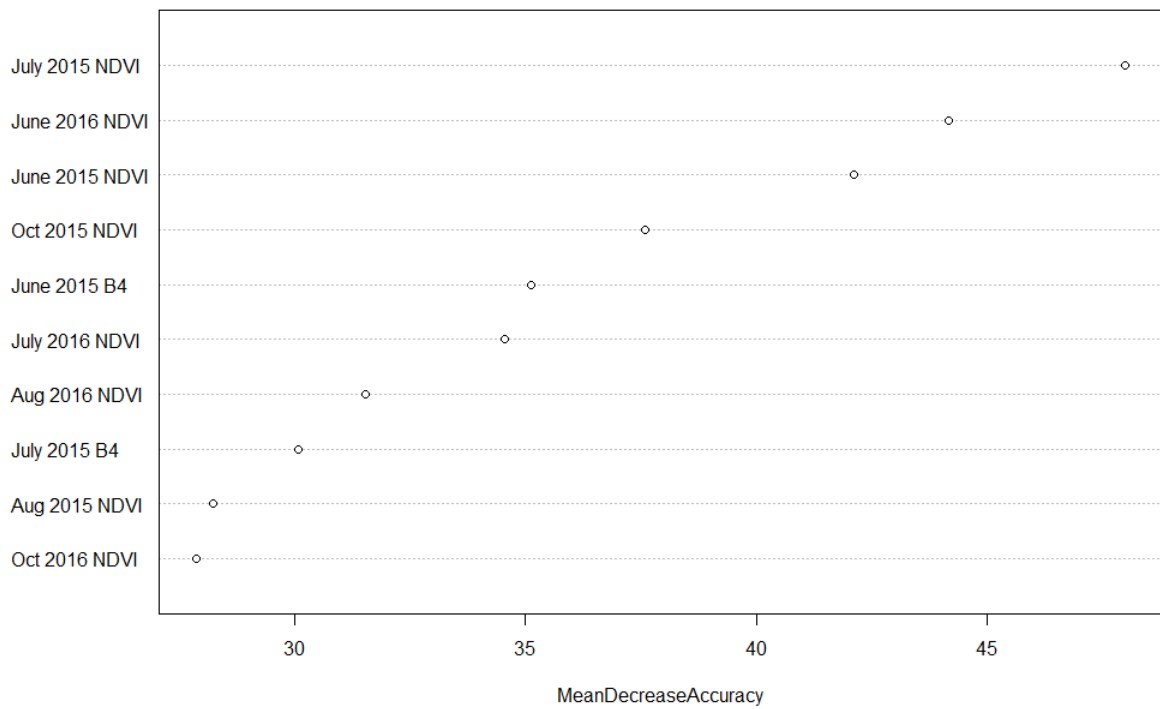


Figure 5: Variable importance reported as mean decrease in accuracy. Ten most important
variables are shown, with B1 and B4 corresponding to spectral bands 1 and 4 respectively from
Pleiades image.  Variable importance is determined by the model output as the decrease in
accuracy due to the exclusion of that variable during the out of bag error calculation process.
Higher mean decrease in accuracy variables are more important in classifying the data.

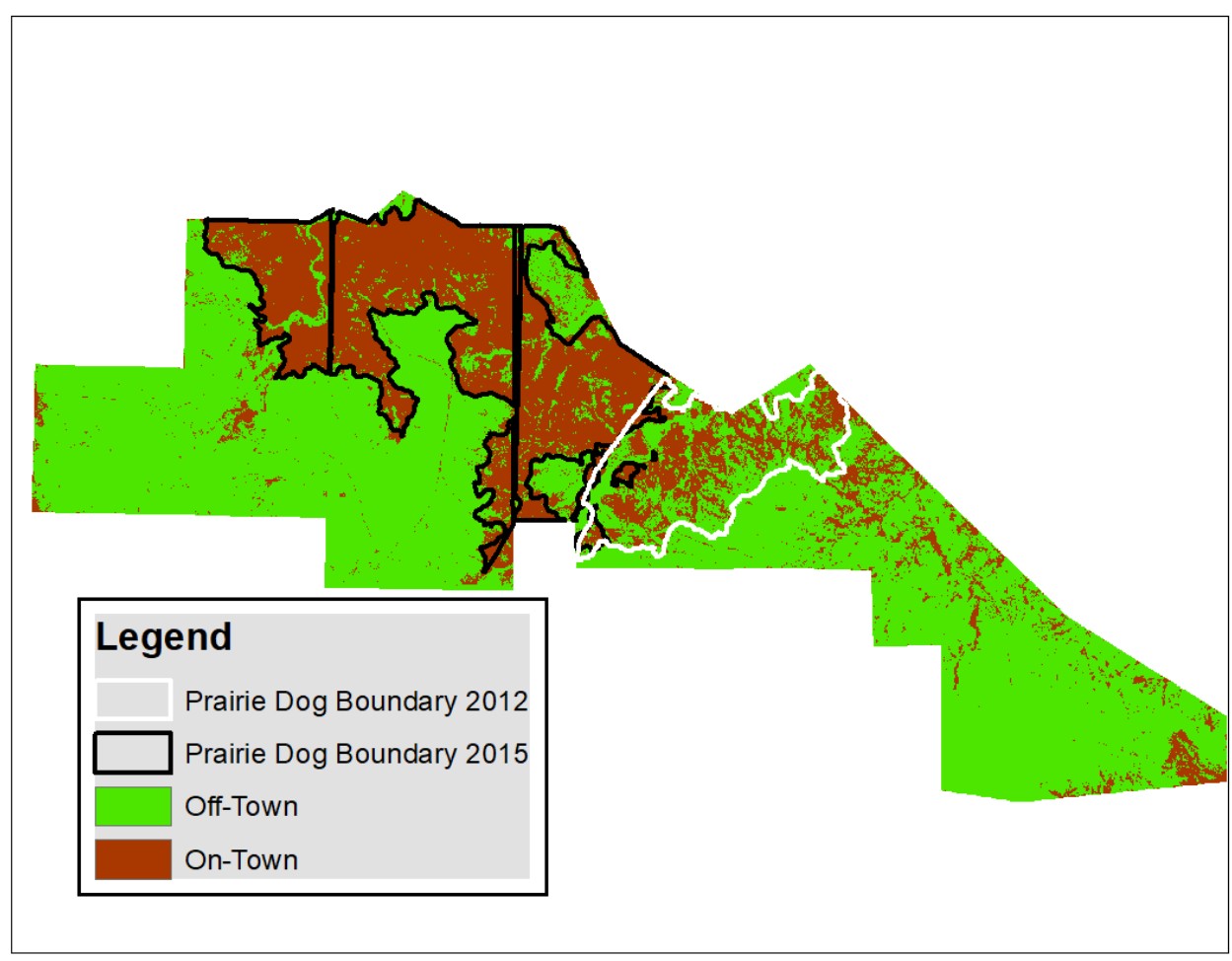

Figure 6: Random forest classification map created from predictions from the combined 2015 and 2016 models.  Off-town areas were created by combining the predicted off-town plant communities (Off-Cool, Off-Warm, and Off-Snow) and on-town plant communities (On-Grass and On-Forb).  The prairie dog boundary was mapped using a handheld GPS unit, the outlined 2012 prairie dog boundary was former prairie dog colony poisoned in 2013.