# Peer review of "Comparing Stability in Random Forest Models to Map Northern Great Plains Plant Communities"

_Biogeosciences, 2019_

## Referee Comment (RC1) · Anonymous Referee #1 · 14 Jul 2019

Overview

This manuscript describes how vegetation groups switch between years based on random forest classification models utilizing remote sensing imagery. The authors demonstrate how to produce highly accurate images with purely spectrally based predictors, and also quantify the variability in their vegetation groups between years. Understanding these shifts is an important undertaking for ecology and remote sensing, however there are several factors that lead to confusion in the interpretation of the results. In addition, the application of Random Forests is limited compared to what is set out in the intro, and when coupled with what seem like arbitrary (or unexplained) decisions

in the approach, I feel their objectives have not been fully met. Specifically the confusion between community types and functional groups needs to be addressed in the manuscript through defining terms, and clarifying the difference between community changes and production differences of functional groups between years. For the assessment of random forests, there is opportunity to develop the analysis much deeper. There are unanswered questions that could be explored with RF models. For example, what number of trees are needed for the model to stabilize, how does the months of imagery (what if I have two instead of five each year) change the classification, why separate models for on and off prairie dog towns (when transitions between these and the three off town types may be important), etc.

Specific Comments

Line 44 – Awkward sentence, and if I understand this correctly, then I disagree. I actually am not surprised by changes in species dominance between years. Composition may stay the same, but representation can change depending on growing season conditions.

Line 48 – Vegetation classification can be done at many scales in multiple vegetation hierarchies, you need to be much more specific here (and throughout) about what you are looking at and where in a vegetation hierarchy your results are relevant.

Line 55 – Very broad and general and probably needs a citation. Take a look at (Browning, D. M., A. Rango, J. W. Karl, C. M. Laney, E. R. Vivoni, and C. E. Tweedie. 2015. Emerging technological and cultural shifts advancing drylands research and management. Frontiers in Ecology and the Environment 13:52-60) to think about how remote sensing fits into monitoring and assessment for rangelands.

Line 90 – You are not exploring "plant and animal interactions" in this paper, although this is the aim of your larger project. This is out of place and confusing. In addition, "plant and animal interactions" is vague, I started thinking about a wealth of LIDAR and similar studies used to create thematic vegetation maps for animal habitat studies. Do

you mean there is limited studies on animal space use across vegetation communities?

Line 101 –Three examples don't prove that RF is better in all situations. And as written, it seems like you cite one study that actually compares RF with other techniques, and this used Landsat, very different than your study. Nothing majorly wrong here, you just need to introduce applications of RF to vegetation classification problems as one useful technique.

Line 114 – Probably this illustrates a limitation in predictor variables, rather than Random Forests. The tool can work at broad scales if the data and processing power is available. See Jones, M. O., B. W. Allred, D. E. Naugle, J. D. Maestas, P. Donnelly, L. J. Metz, J. Karl, R. Smith, B. Bestelmeyer, C. Boyd, J. D. Kerby, and J. D. McIver. 2018. Innovation in rangeland monitoring: annual, 30 m, plant functional type percent cover maps for U.S. rangelands, 1984-2017. Ecosphere 9.

Line 120 – Plant community classification (used here and throughout) can be conducted on many different levels of vegetation hierarchies (like the USNVC). Or you can ask other questions, like changes in productivity between years. Community type generally shifts when a system crosses a threshold from disturbance/stressors, through succession, etc. You need to define much better in this manuscript what you mean, and what you are looking at in regards to, for example, plant community classification vs. plant community species or productivity. A single Landsat image may work very well for some purposes, but for the more detailed questions like yours, multiple images may be required (although you did not actually test accuracy differences between the number of images). You seem to be focused more on classification of functional groups rather than a community.

Line 121 – The three references here are specific studies (two over 10 years old), not reviews of plant classification studies. Maybe in the past it was more common to use a single time period, but seems now with increased computing power and the availability of the entire Landsat achieve, etc., it is very common for much more robust and multiple

acquisition studies. Maybe phrase this more to acknowledge this evolution. Line 135 – Again, what is a plant community. Have you really found or are looking that the community changes? Or are you looking at species representation within a community, i.e. functional group dominance and shifts in this between years.

Line 143 – NGP probably too broad for the implications of your study. Be more specific with the MLRA, or mixed grass prairie systems, etc. that you are testing.

Intro – Lots of general and vague statements in here and limited citations supporting broad brush statements. I suggest going through the intro to make it more specific. For example, landscape, local, various etc. scales will mean different things to every reader. Define these or what they are for the studies you cite. Also make sure your statements are supported by citations or explained. For example, line 61 – 65 is not a summation or conclusion of the paragraph, so these new statements should be supported. Finally, you should mention this is part of a larger study looking at cattle use compared to prairie dog prevalence and impact to pastures, but the paragraph starting on line 130 had me confused between what this study was going to do, and what the larger study did.

Line 174 – These sound more like plant functional groups than communities. Nothing wrong with mapping those, but the terminology issues are prevalent and I believe confuse your conclusions. Changes in representation are common between years, changes in community are a different boat.

Line 199 – Why not compare them all together? You need to add rationale for why separating these out beforehand is appropriate. If you want to scale up your study, how will you separate out prairie dog towns at the "landscape" scale (watershed, county, etc.). As another option, I would find this much more compelling if the comparisons and RF models were tested to separate out all five groups. This would be a much more thorough test of RF.

Line 217 – Do you think the wider spectral bands (compared to Landsat or UAV options

available) played into your results at all?

Line 227 – Again, this needs more justification then saying they are mutually exclusive. You either decided to map prairie dog towns separately from the rest of the study area (which you need to justify why) or you could test what the implications are of not having mapped towns in the first place (which also can vary between years).

Line 239 – Why only 100, when the default is higher (which is used for the number of nodes)? You may be ok here, but in many cases, at this point the model error is just beginning to stabilize. You could examine the impact of the number of trees on your model by looking over a range of "number of tree" values.

Line 241 – Why just spectral bands as input into the models? You don't explicitly say your objective is to use just satellite imagery (and prior to the RF algorithm you used other data, e.g. to differentiated prairie dog towns and off site)

Line 246 – How did you apply your models to produce predictions for prairie dog towns vs. off town locations? I think you run the predictions on two separate parts of the study area (be explicit).

Line 262 – The way your "communities" were picked seem to almost guarantee this? You picked areas "dominated" by three (or two) very different functional groups. Is this overlap more than you expected, and what is the overlap? This very much may help explain the differences between years.

Line 287 – Are the models unstable, or does this indicate the models are accurate within years, but species representation (as seen through your methods) changes between years in heterogenous areas?

Line 303 – These peaks seem like they very much may affect the production of warm vs. cool season grasses between the years as well.

Line 304 – Was there a temperature difference between years as well? These curves seem farther apart then I would expect just based on precip

Line 328 – Is there a transition zone at the edge of the prairie dog towns too?

Line 340 – Based on your discussion so far, what is a more accurate thematic map? Which year is the truth, if the heterogeneous transition zones may switch categories depending on which group dominates in a given year? How about comparing this map to the two yearly maps?

Line 351 – Any limitations in the approach though? How about the lack of coefficients for your variables? I.e. good for prediction, not as good for understanding relationships

Line 353 – Why not include the variable importance for the combined model?

Line 562 – Break this out to be more specific on the changes per year (what was it in 2015 and what is it now in 2016) rather than lumping the switches between types that switch both ways in the two years. If there is a dominant pattern of switch that would be useful for your conclusions.

Line 588 – How are the draws mapped? These are not one of your groups, need to talk about this in the methods.

Technical Corrections

Line 3 – Consider replacing stability, I think this could be confused with other definitions and is not quite what you mean

Line 32 – Replace highly with high

Line 46 – Replace instability with disagreement

Line 66 – Remove colonization and replace dog with dogs

Line 86 – First time you use the acronym NDVI. Write out fully.

Line 98 – Replace several with many (or similar idea)

Line 101 – Replace proven with demonstrated

Line 109 – There are a lot more RF packages and implementation options now, compared to 2013. Standard software like R, ERDAS Imagine, QGIS, and ArcGIS have RF, as well as more specialized options like Ecognition (and even Google Earth Engine). I don't think you need this sentence, not relevant to the paper.

Line 173 – Need year you accessed the Mesonet data

Line 181 – About how big are these (median, range, etc.)

Line 192 – How were they randomly located?

Line 236 – Did you consider other potential predictors that you could derive from these inputs?

Line 239 – What is the default number of nodes. Define this.

Line 256 – A table of the species for each of the five groups would really help. Would also help understand what "dominated" means for your training sites.

Line 267 – Mishra and Crews should be outside parentheses

Line 310 – What was the 2014 precip then? Dry?

Line 355 – For the town or off-site model?

---

## Referee Comment (RC2) · Anonymous Referee #2 · 17 Sep 2019

The authors use the unique plant community signature of Prairie Dog colonies to challenge RF methods, but the novelty of this approach is never articulated. Explain early on, with references, why temporal and spatial characteristics of prairie dog influence on vegetation makes it an interesting challenge for remote sensing and the combined ecological/rangeland management/remote sensing triumvirate of the manuscript will be clearer to the reader. The Introduction needs to be restructured and I recommend the Results and Discussion be entirely re-written, it was extremely difficult to follow and all of the cool aspects of this interesting study were either buried or not mentioned at all. After rather major revisions I can see how this paper could be acceptable for publication. It is technically sound for the most part but needs major changes.

[Figure]

Minor comments: The ecological justification for investigating Prairie Dog towns was somewhat lacking in the abstract. Is this study fundamentally about identifying colonies from remote platforms or using prairie dog colonies as an interesting opportunity to advance statistical techniques in remote sensing?

The statement on line 43 is somewhat fuzzy. The cautious note at the end of the abstract is forthcoming.

The transition from line 65 to 66 is a bit harsh. The narrative 'funnels' from remote sensing in general to prairie dog colonies in particular far too rapidly. As a consequence, the reader is left wondering if the central theme is prairie dog colony identification or remote sensing techniques or rangeland and cattle management (or all of the above, and if so how do they fit together).

The paragraph beginning line 79 is 'listy' and reads like a few random manuscripts that the authors read. How do these fit together to advance the overall objective of the study? I recommend restructuring the Introduction. 'Writing Science' by Schimel is a good text for describing logical flow in scientific manuscripts.

From the paragraph on line 101 it appears that the objective isn't to compare RF against different techniques, which is fine. But the opportunity to use the subtle (or not so subtle) vegetation changes induced by prairie dog colonies to challenge RF methods isn't brought to the forefront. This is a missed opportunity in my opinion. Note also in line 146 that a goal could also be to investigate prairie dog and plant ecology: you don't always have to bring it back to cattle foraging. The Utah and Mexican Prairie Dogs are endangered after all.

156: The Ecological Sites notion was new to me and the descriptions sound like soil types. Are these a USDA thing?

162: I'm confused, I always thought that Kentucky bluegrass was Poa pratensis.

173: the temperature and precip measurements are great but please specify the

mesonet used (South Dakota).

174: using common abbreviations like 'pdf' or common words like 'snow', 'cool', and 'warm' will lead to confusion. Sites are either on towns or off, so using PD with subscript f or g, then O (or similar, even 'NPD' as used on line 201 without previous description) with subscripted snowberry, c3, and c4 would help me at least. There is a lot to digest here and making things easier for the reader can go a long way.

I'm not entirely sure why an ordination, MRPP, NMS, Bray-Curtis, etc. was used for pre-defined vegetation types. Weren't they already selected to be different from each other? Is the point of this analysis to guarantee that the five vegetation types are in fact different from each other (e.g. line 256)? In this case of course it's fine to do so.

NDVI probably doesn't need to be defined on 231 although a note about any differences in the spectral resolution of the red and NIR among Pleiades and other common satellites may be interesting for the Discussion.

276 is probably a methods point and 278 may even be an Introduction point. Literature as a whole needs to be woven into the narrative. In general, any time a sentence starts with the author of a paper, the sentence needs to be changed. Doing this makes the author(s) the subject(s) of the sentence. The topic at hand should be the topic of the sentence. Please start a sentence with authors only when those authors are the subject of the sentence, which can happen.

The paragraph beginning 265 could benefit from a few more quantitative values rather than qualitative ones like 'high degree' and 'lower'.

296: I disagree somewhat. Different species will be more prominent during different times of the year (e.g. cool vs warm season grasses).

The manuscript would probably benefit from separating the results and discussion to show first what happened then explain it. The discussion never comes back to prairie dogs.

Please make font sizes larger in the figures. They are often hard to read.

From Fig. 5 and 6 it appears that prairie dog colonies, at least in this area of SD, can be identified with a relatively large degree of accuracy. This needs to be made more prominent in the discussion.

———————————————

---

## Author Comment (AC1) · 15 Oct 2019

Overview This manuscript describes how vegetation groups switch between years based on random forest classification models utilizing remote sensing imagery. The authors demonstrate how to produce highly accurate images with purely spectrally based predictors, and also quantify the variability in their vegetation groups between years. Understanding these shifts is an important undertaking for ecology and remote sensing, however there are several factors that lead to confusion in the interpretation of the results. In addition, the application of Random Forests is limited compared to what is set out in the intro, and when coupled with what seem like arbitrary (or unexplained) decisions, I feel their objectives have not been fully met. Specifically the confusion between community types and functional groups needs to be addressed in the manuscript through defining terms, and clarifying the difference between community changes and production differences of functional groups between years. For the assessment of random forests, there is opportunity to develop the analysis much deeper. There are unanswered questions that could be explored with RF models. For example, what number of trees are needed for the model to stabilize, how does the months of imagery (what if I have two instead of five each year) change the classification, why separate models for on and off prairie dog towns (when transitions between these and the three off town types may be important), etc. Specific Comments

-There are several comments by this reviewer regarding definition of plant community as used in this paper. Plant communities can be described at many scales; at a very large scale, we could simply distinguish between grassland and shrubland communities, knowing full well that there is tremendous potential for variability within those two categories. We could also describe plant communities at a very fine scale, combining land units only when the percent compositions (either by cover or biomass) of the communities are nearly identical. For the purpose of the overall study within which this study was a part, we determined that an intermediate scale provided reasonable distinctions between areas on the landscape without being too broad or too detailed to be useful. Thus we combined the plant communities into categories defined by several functional groups. As indicated, above, we are very cognizant of the fact that plant communities defined by a dominant functional group will not only have other functional groups within those communities, but the composition of those other functional groups may vary widely. We will add an explanation early in the manuscript that details the scale at which we are dealing with plant communities to clarify.

-Regarding the issue of number of trees to attain stability, prior remote sensing research has focused on differing number of trees on model predictions, as well as seasonality of imagery collection on classification error within the same year. While our

data could be used to explore that question, that is not the focus of this paper.

Line 44 – Awkward sentence, and if I understand this correctly, then I disagree. I actually am not surprised by changes in species dominance between years. Composition may stay the same, but representation can change depending on growing season conditions.

-Sentence can be re-written for clarification. Grasslands within the northern great plains are primarily composed of perennial species, thus it is unlikely that shifts in dominance will occur from one year to the next. Substantive changes to the relative dominance of species in plant communities tend to occur over longer time spans. It is more likely that, when plant communities dominated by one functional group (e.g. C3 grasses) also have a fairly large percentage composition of plants from a different functional group (e.g. C4 grasses), specific growing conditions may result in the appearance of a dominance shift.

Line 48 – Vegetation classification can be done at many scales in multiple vegetation hierarchies, you need to be much more specific here (and throughout) about what you are looking at and where in a vegetation hierarchy your results are relevant.

-Sentence can be re-written to specify high resolution satellite imagery for northern great plains plant communities.

Line 55 – Very broad and general and probably needs a citation. Take a look at (Browning, D. M., A. Rango, J. W. Karl, C. M. Laney, E. R. Vivoni, and C. E. Tweedie. 2015. Emerging technological and cultural shifts advancing drylands research and management. Frontiers in Ecology and the Environment 13:52-60) to think about how remote sensing fits into monitoring and assessment for rangelands.

-Citation can be added.

Line 90 – You are not exploring "plant and animal interactions" in this paper, although this is the aim of your larger project. This is out of place and confusing. In addition,

"plant and animal interactions" is vague, I started thinking about a wealth of LIDAR and similar studies used to create thematic vegetation maps for animal habitat studies. Do you mean there is limited studies on animal space use across vegetation communities?

-The aim of the larger project is to explore plant-animal interactions, specifically livestock use of plant communities on the landscape. However, prairie dogs have a large impact on plant communities and can drive differences in species composition, production, etc. There are certainly studies on animal selection of vegetation communities; the use of thematic maps developed from high resolution satellite imagery is, however, much less studied.

Line 101 –Three examples don't prove that RF is better in all situations. And as written, it seems like you cite one study that actually compares RF with other techniques, and this used Landsat, very different than your study. Nothing majorly wrong here, you just need to introduce applications of RF to vegetation classification problems as one useful technique.

-RF may not be the best algorithm for all situations. The studies in this paragraph highlight some of the uses of RF to classify plant communities. Additional studies can be added to demonstrate RF outperforming other methods to demonstrate the high degree of accuracy for these models. We can make a change from "proven" to "demonstrated" in the sentence to alleviate the concern.

Line 114 – Probably this illustrates a limitation in predictor variables, rather than Random Forests. The tool can work at broad scales if the data and processing power is available. See Jones, M. O., B. W. Allred, D. E. Naugle, J. D. Maestas, P. Donnelly, L. J. Metz, J. Karl, R. Smith, B. Bestelmeyer, C. Boyd, J. D. Kerby, and J. D. McIver. 2018. Innovation in rangeland monitoring: annual, 30 m, plant functional type percent cover maps for U.S. rangelands, 1984-2017. Ecosphere 9.

-The Juel et al study mentioned deals with high resolution imagery, compared to 30m Landsat data as mentioned in the suggested study. Perhaps the issue of spatial trans-
ferability of models becomes a greater issue at fine scale mapping.

Line 120 – Plant community classification (used here and throughout) can be conducted on many different levels of vegetation hierarchies (like the USNVC). Or you can ask other questions, like changes in productivity between years. Community type generally shifts when a system crosses a threshold from disturbance/stressors, through succession, etc. You need to define much better in this manuscript what you mean, and what you are looking at in regards to, for example, plant community classification vs. plant community species or productivity. A single Landsat image may work very well for some purposes, but for the more detailed questions like yours, multiple images may be required (although you did not actually test accuracy differences between the number of images). You seem to be focused more on classification of functional groups rather than a community.

-See initial response, above, to the overview comments. Though several of our plant communities are dominated by specific functional groups (shrubs vs grass or grass vs forb), this does not mean, nor did we intend it to mean, these are the only functional groups within those communities. Communities are composed of a variety of species representing multiple functional groups. For example, forb dominated sites will also contain grass species and vice versa. We are classifying plant communities in this study based on the dominant functional groups within a plant community.

Line 121 – The three references here are specific studies (two over 10 years old), not reviews of plant classification studies. Maybe in the past it was more common to use a single time period, but seems now with increased computing power and the availability of the entire Landsat achieve, etc., it is very common for much more robust and multiple acquisition studies. Maybe phrase this more to acknowledge this evolution.

-Sentenced can be changed to acknowledge this.

Line 135 – Again, what is a plant community. Have you really found or are looking that the community changes? Or are you looking at species representation within a

community, i.e. functional group dominance and shifts in this between years.

-See initial response, above, to the overview comments. A plant community is a collection of species within an area of a relatively uniform composition different from neighboring patches. In this study, differences in neighboring patches are sometime evident in differences in dominant functional group (forb vs grass) or differences in photosynthetic pathways (C3 vs C4 grasses). The aim of this study is not to measure community changes, as shifts in plant communities occur over longer time scales within perennial northern great plains plant communities.

Line 143 – NGP probably too broad for the implications of your study. Be more specific with the MLRA, or mixed grass prairie systems, etc. that you are testing. Intro – Lots of general and vague statements in here and limited citations supporting broad brush statements. I suggest going through the intro to make it more specific. For example, landscape, local, various etc. scales will mean different things to every reader. Define these or what they are for the studies you cite. Also make sure your statements are supported by citations or explained. For example, line 61 – 65 is not a summation or conclusion of the paragraph, so these new statements should be supported. Finally, you should mention this is part of a larger study looking at cattle use compared to prairie dog prevalence and impact to pastures, but the paragraph starting on line 130 had me confused between what this study was going to do, and what the larger study did.

-Intro can be changed to include mixed grass prairie specifically, though much of the Northern Great Plains is comprised of mixed grass prairie. Intro can specify the differences in scales between citations such as Landsat imagery versus high resolution imagery. Line 130 states that this is part of a larger study linking livestock use of plant communities within pastures occupied by prairie dogs. Line 130 highlights the larger study, then shifts to the specific goals of this paper.

Line 174 – These sound more like plant functional groups than communities. Nothing wrong with mapping those, but the terminology issues are prevalent and I believe confuse your conclusions. Changes in representation are common between years, changes in community are a different boat.

-See initial response, above, to the overview comments. In this study, plant communities are organized by dominant functional groups, with the knowledge that other functional groups are also present within these communities.

Line 199 – Why not compare them all together? You need to add rationale for why separating these out beforehand is appropriate. If you want to scale up your study, how will you separate out prairie dog towns at the "landscape" scale (watershed, county, etc.). As another option, I would find this much more compelling if the comparisons and RF models were tested to separate out all five groups. This would be a much more thorough test of RF.

-Separate models can be created to test the ability of RF to separate all five groups. The study was setup to map plant communities of interest on- and off-town for the purpose of relating this data to livestock use. Though plant communities are defined by dominant functional group, functional groups are not mutually exclusive within those communities, however, a site can only occur on- or off-town and not both thus is mutually exclusive.

Line 217 – Do you think the wider spectral bands (compared to Landsat or UAV options) played into your results at all?

-No

Line 227 – Again, this needs more justification then saying they are mutually exclusive. You either decided to map prairie dog towns separately from the rest of the study area (which you need to justify why) or you could test what the implications are of not having mapped towns in the first place (which also can vary between years).

-Prairie dog towns were mapped separately from the rest of the study area as part

of the larger project objectives. An additional test can be whether plant communities on prairie dog towns are different enough to be remotely sensed from off-town areas. Though the scope of this study was to test the ability to map different plant communities on and off town (two separate areas separated by the presence/absence of prairie dogs) and assess differences in RF model predictions between years.

Line 239 – Why only 100, when the default is higher (which is used for the number of nodes)? You may be ok here, but in many cases, at this point the model error is just beginning to stabilize. You could examine the impact of the number of trees on your model by looking over a range of "number of tree" values.

-100 trees were enough to allow the model error to stabilize. Adding additional trees (default=500) was computationally prohibitive.

Line 241 – Why just spectral bands as input into the models? You don't explicitly say your objective is to use just satellite imagery (and prior to the RF algorithm you used other data, e.g. to differentiated prairie dog towns and off site)

-It is explicitly mentioned in the goals of the project to 'assess the utility of using a RF model with high resolution satellite imagery to classify plant communities'. Delineating prairie dog town boundaries with GIS would be akin to outlining the boundary of any other study area of interest. Line 246 – How did you apply your models to produce predictions for prairie dog towns vs. off town locations? I think you run the predictions on two separate parts of the study area (be explicit). The 'predict' function in Program R was used to apply the models. For example 2015 on town model was used to predict class belonging to on-town areas.

Line 262 – The way your "communities" were picked seem to almost guarantee this? You picked areas "dominated" by three (or two) very different functional groups. Is this overlap more than you expected, and what is the overlap? This very much may help explain the differences between years.

-We would expect a large separation in ordination space based on how plant communities were selected. I think it is of value for plant classification studies to demonstrate that the plant communities one is classifying are actually distinct. The amount of overlap between plant communities may also factor into error rates or help explain differences between years. As mentioned prior, plant communities may be dominated by a specific functional groups, but other functional groups and species exist within these areas.

Line 287 – Are the models unstable, or does this indicate the models are accurate within years, but species representation (as seen through your methods) changes between years in heterogenous areas?

-Unsure what is meant by species representation. Given that these are perennial plant communities, it would be unlikely for major shifts in dominance to occur between successive years without a major disturbance event (i.e. fire).

Line 303 – These peaks seem like they very much may affect the production of warm vs. cool season grasses between the years as well.

-Agreed, it could affect production to some extent.

Line 304 – Was there a temperature difference between years as well? These curves seem farther apart then I would expect just based on precip.

-Temperature was similar between years, additional weather data could be added. Prior research has shown annual above ground primary production is related to current as well as previous two years precipitation. The above average rainfall at the study site in 2015 could have added to the increase in average NDVI in 2016 when compared to 2015 through an increase in cumulative biomass or production at the site.

Line 328 – Is there a transition zone at the edge of the prairie dog towns too?

-Though there are transition zones at the edge of prairie dog towns, they tend to be much sharper boundaries and occur in off-town sites. This results in much more distinct

boundaries and improves the ease of mapping colonies.

Line 340 – Based on your discussion so far, what is a more accurate thematic map? Which year is the truth, if the heterogeneous transition zones may switch categories depending on which group dominates in a given year? How about comparing this map to the two yearly maps?

-The map which includes both 2015 and 2016 data is likely the most accurate map, as demonstrated in the lower error rates. More information (spectral values across seasons and years) would produce a more accurate thematic map. As mentioned prior, switching in dominance, especially functional group dominance, between consecutive years is unlikely to occur in perennial mixed-grass prairie ecosystems without a major disturbance occurring.

Line 351 – Any limitations in the approach though? How about the lack of coefficients for your variables? I.e. good for prediction, not as good for understanding relationships

-The goal of creating predictive models is to generate good predictions. The aim of this study was prediction, not inference.

Line 353 – Why not include the variable importance for the combined model?

-This can be included.

Line 562 – Break this out to be more specific on the changes per year (what was it in 2015 and what is it now in 2016) rather than lumping the switches between types that switch both ways in the two years. If there is a dominant pattern of switch that would be useful for your conclusions.

-The table can be broken out to include specific changes between the years.

Line 588 – How are the draws mapped? These are not one of your groups, need to talk about this in the methods. Technical Corrections Line 3 – Consider replacing stability, I think this could be confused with other definitions and is not quite what you mean

-Draws were delineated visually via ArcGIS and imagery. This can be included in the methods.

Line 32 – Replace highly with high Line 46 – Replace instability with disagreement Line 66 – Remove colonization and replace dog with dogs

-Can be included.

Line 86 – First time you use the acronym NDVI. Write out fully.

-Can be included.

Line 98 – Replace several with many (or similar idea) -Can be included.

Line 101 – Replace proven with demonstrated -Can be included.

Line 109 – There are a lot more RF packages and implementation options now, compared to 2013. Standard software like R, ERDAS Imagine, QGIS, and ArcGIS have RF, as well as more specialized options like Ecognition (and even Google Earth Engine). I don't think you need this sentence, not relevant to the paper. -Can be deleted.

Line 173 – Need year you accessed the Mesonet data Line 181 – About how big are these (median, range, etc.) -Can be included.

Line 192 – How were they randomly located? -Plot frames were tossed into the area of interest to determine sampling area.

Line 236 – Did you consider other potential predictors that you could derive from these inputs?

-We didn't include additional vegetation indices from the spectral data. Additional metrics could include elevation, ecological site, etc. though given the low training error from the models this would not likely change the predictions.

Line 239 – What is the default number of nodes. Define this.

-500

Line 256 – A table of the species for each of the five groups would really help. Would also help understand what "dominated" means for your training sites. -Could be included.

Line 267 – Mishra and Crews should be outside parentheses -Can be changed.

Line 310 – What was the 2014 precip then? Dry? -Wet

Line 355 – For the town or off-site model? -Off town, can also be included.

---

## Author Comment (AC2) · 15 Oct 2019

The authors use the unique plant community signature of Prairie Dog colonies to challenge RF methods, but the novelty of this approach is never articulated. Explain early on, with references, why temporal and spatial characteristics of prairie dog influence on vegetation makes it an interesting challenge for remote sensing and the combined ecological/rangeland management/remote sensing triumvirate of the manuscript will be clearer to the reader. The Introduction needs to be restructured and I recommend the Results and Discussion be entirely re-written, it was extremely difficult to follow and all of the cool aspects of this interesting study were either buried or not mentioned

at all. After rather major revisions I can see how this paper could be acceptable for publication. It is technically sound for the most part but needs major changes.

Minor comments: The ecological justification for investigating Prairie Dog towns was somewhat lacking in the abstract. Is this study fundamentally about identifying colonies from remote platforms or using prairie dog colonies as an interesting opportunity to advance statistical techniques in remote sensing?

-The study was part of a larger study focusing on livestock production and diet selection within pastures occupied by prairie dog towns. A component of that study was to use remote sensing to identify plant communities of interest within the pastures, and explore how well RF models perform using imagery from different years.

The statement on line 43 is somewhat fuzzy. The cautious note at the end of the abstract is forthcoming.

-Sentence can be amended for clarity.

The transition from line 65 to 66 is a bit harsh. The narrative 'funnels' from remote sensing in general to prairie dog colonies in particular far too rapidly. As a consequence, the reader is left wondering if the central theme is prairie dog colony identification or remote sensing techniques or rangeland and cattle management (or all of the above, and if so how do they fit together).

-This aligns with the overarching goal of the larger project, linking plant communities' on- and off-town to livestock behavior. As stated at the end of the paragraph, understanding these dynamics requires the ability to map plant communities at landscape scales.

The paragraph beginning line 79 is 'listy' and reads like a few random manuscripts that the authors read. How do these fit together to advance the overall objective of the study? I recommend restructuring the Introduction. 'Writing Science' by Schimel is a good text for describing logical flow in scientific manuscripts.

-These manuscripts fit into the overall objective of the study by demonstrating mapping of various plant communities using spectrally derived data from satellite imagery, which is the overall goal of this study.

From the paragraph on line 101 it appears that the objective isn't to compare RF against different techniques, which is fine. But the opportunity to use the subtle (or not so subtle) vegetation changes induced by prairie dog colonies to challenge RF methods isn't brought to the forefront. This is a missed opportunity in my opinion. Note also in line 146 that a goal could also be to investigate prairie dog and plant ecology: you don't always have to bring it back to cattle foraging. The Utah and Mexican Prairie Dogs are endangered after all.

-Numerous studies have investigated prairie dog impacts on plant ecology. As mentioned in the paragraph beginning on line 66, studies have demonstrated that prairie dogs can have a large impact plant species composition, and older core areas often become characterized by annual forbs species and low production, which can directly impact livestock production. Though other species of prairie dog in the west are endangered, black tailed prairie dogs in the northern great plains are not federally listed and are often the focus of debates over biodiversity, conservation, and agricultural production. In addition, the rangelands in the Northern Great Plains are primarily in private ownership. Thus the considerable overlap between cattle and prairie dogs is very important.

156: The Ecological Sites notion was new to me and the descriptions sound like soil types. Are these a USDA thing?

-Ecological sites are used heavily by USDA and USDA agencies as well as by private land managers. They are a distinctive type of land with specific soil and physical characteristics produce unique vegetation.

162: I'm confused, I always thought that Kentucky bluegrass was Poa pratensis.

-Kentucky bluegrass is Poa pratensis, this will be corrected.

173: the temperature and precip measurements are great but please specify the mesonet used (South Dakota).

-This can be added.

174: using common abbreviations like 'pdf' or common words like 'snow', 'cool', and 'warm' will lead to confusion. Sites are either on towns or off, so using PD with subscript f or g, then O (or similar, even 'NPD' as used on line 201 without previous description) with subscripted snowberry, c3, and c4 would help me at least. There is a lot to digest here and making things easier for the reader can go a long way.

-Acronyms can be changed to improve clarity for the reader.

I'm not entirely sure why an ordination, MRPP, NMS, Bray-Curtis, etc. was used for pre-defined vegetation types. Weren't they already selected to be different from each other? Is the point of this analysis to guarantee that the five vegetation types are in fact different from each other (e.g. line 256)? In this case of course it's fine to do so.

-We would expect a large separation in ordination space based on how plant communities were selected. I think it is of value for plant classification studies to demonstrate that the plant communities one is classifying are actually distinct. The amount of overlap between plant communities may also factor into error rates or help explain differences between years.

NDVI probably doesn't need to be defined on 231 although a note about any differences in the spectral resolution of the red and NIR among Pleiades and other common satellites may be interesting for the Discussion.

-NDVI definition can be removed.

276 is probably a methods point and 278 may even be an Introduction point. Literature as a whole needs to be woven into the narrative. In general, any time a sentence

starts with the author of a paper, the sentence needs to be changed. Doing this makes the author(s) the subject(s) of the sentence. The topic at hand should be the topic of the sentence. Please start a sentence with authors only when those authors are the subject of the sentence, which can happen.

-Respectfully disagree. Listing an author at the beginning of a sentence is a common convention in ecological literature.

The paragraph beginning 265 could benefit from a few more quantitative values rather than qualitative ones like 'high degree' and 'lower'.

-Tables are referenced giving specific values. Most ecological and agricultural publications require that specific values be either in tables or in the text, but not both.

296: I disagree somewhat. Different species will be more prominent during different times of the year (e.g. cool vs warm season grasses).

-I agree somewhat that different species will be more prominent during different times of the year. This is especially true of forb species that flush early in the season in the region. However substantial increases or decreases in perennial cool and warm season grasses from one season to the next are rare without a major disturbance. They often occur in mixed stands, and thus occupy the same physical space on the landscape. Thus changes in pixels from warm season to cool season (and the reverse) is less likely due to a real shift in composition and more likely due to phenological responses to climatic variables.

The manuscript would probably benefit from separating the results and discussion to show first what happened then explain it. The discussion never comes back to prairie dogs.

-Respectfully disagree. Separation vs. combination of the results and the discussion is often a matter of preference, however combination is very common in the literature. Regarding prairie dogs, the objective of this paper is not to focus on prairie dogs or

cattle; it is to focus on the use of RF in distinguishing between the associated plant communities.

Please make font sizes larger in the figures. They are often hard to read. -This can be changed.

From Fig. 5 and 6 it appears that prairie dog colonies, at least in this area of SD, can be identified with a relatively large degree of accuracy. This needs to be made more prominent in the discussion.

-Though the objective of this study was not to remotely sense prairie dog colonies, the use of satellite imagery to accomplish this goal can be mentioned in the discussion and may warrant further investigation.

---

## Author Response (AR1)

**Response to Reviewer 1**

Overview This manuscript describes how vegetation groups switch between years based on random forest classification models utilizing remote sensing imagery. The authors demonstrate how to produce highly accurate images with purely spectrally based predictors, and also quantify the variability in their vegetation groups between years. Understanding these shifts is an important undertaking for ecology and remote sensing, however there are several factors that lead to confusion in the interpretation of the results. In addition, the application of Random Forests is limited compared to what is set out in the intro, and when coupled with what seem like arbitrary (or unexplained) decisions, I feel their objectives have not been fully met. Specifically the confusion between community types and functional groups needs to be addressed in the manuscript through defining terms, and clarifying the difference between community changes and production differences of functional groups between years. For the assessment of random forests, there is opportunity to develop the analysis much deeper. There are unanswered questions that could be explored with RF models. For example, what number of trees are needed for the model to stabilize, how does the months of imagery (what if I have two instead of five each year) change the classification, why separate models for on and off prairie dog towns (when transitions between these and the three off town types may be important), etc. Specific Comments

There are several comments by this reviewer regarding definition of plant community as used in this paper. Plant communities can be described at many scales; at a very large scale, we could simply distinguish between grassland and shrubland communities, knowing full well that there is tremendous potential for variability within those two categories. We could also describe plant communities at a very fine scale, combining land units only when the percent compositions (either by cover or biomass) of the communities are nearly identical. For the purpose of the overall study within which this study was a part, we determined that an intermediate scale provided reasonable distinctions between areas on the landscape without being too broad or too detailed to be useful. Thus we combined the plant communities into categories defined by several functional groups.

To add clarity to the manuscript, we added detailed information on how we defined plant communities for this study.

Regarding the issue of number of trees to attain stability, the number of trees was increased to 200, and plots were generated to check for error stabilization (typically around 50 trees). This has been included in the manuscript. Prior remote sensing research has focused on seasonality of imagery collection on classification error within the same year. While our data could be used to explore that question, that is not the focus of this paper.

Additional changes in the manuscript include combining all plant communities into the RF model instead of separating them out to satisfy the reviewers.

Line 44 – Awkward sentence, and if I understand this correctly, then I disagree. I actually am not surprised by changes in species dominance between years. Composition may stay the same, but representation can change depending on growing season conditions.

Sentence has been removed

Line 48 – Vegetation classification can be done at many scales in multiple vegetation hierarchies, you need to be much more specific here (and throughout) about what you are looking at and where in a vegetation hierarchy your results are relevant.

Scale of studies has been included.

Line 55 – Very broad and general and probably needs a citation. Take a look at (Browning, D. M., A. Rango, J. W. Karl, C. M. Laney, E. R. Vivoni, and C. E. Tweedie. 2015. Emerging technological and cultural shifts advancing drylands research and management. Frontiers in Ecology and the Environment 13:5260) to think about how remote sensing fits into monitoring and assessment for rangelands.

Citation was added.

Line 90 – You are not exploring "plant and animal interactions" in this paper, although this is the aim of your larger project. This is out of place and confusing. In addition, "plant and animal interactions" is vague, I started thinking about a wealth of LIDAR and similar studies used to create thematic vegetation maps for animal habitat studies. Do you mean there is limited studies on animal space use across vegetation communities?

The aim of the larger project is to explore plant-animal interactions, specifically livestock use of plant communities on the landscape.  However, prairie dogs have a large impact on plant communities and can drive differences in species composition, production, etc.  This has been included in the introduction as an area of focus.

Line 101 –Three examples don't prove that RF is better in all situations. And as written, it seems like you cite one study that actually compares RF with other techniques, and this used Landsat, very different than your study. Nothing majorly wrong here, you just need to introduce applications of RF to vegetation classification problems as one useful technique.

RF may not be the best algorithm for all situations.  This paragraph has been changed to highlight the utility of RF in remote sensing using different scales of imagery.

Line 114 – Probably this illustrates a limitation in predictor variables, rather than Random Forests. The tool can work at broad scales if the data and processing power is available. See Jones, M. O., B. W. Allred, D. E. Naugle, J. D. Maestas, P. Donnelly, L. J. Metz, J. Karl, R. Smith, B. Bestelmeyer, C. Boyd, J. D. Kerby, and J. D. McIver. 2018. Innovation in rangeland monitoring: annual, 30 m, plant functional type percent cover maps for U.S. rangelands, 1984-2017. Ecosphere 9.

The Juo study mentioned deals with high resolution imagery, compared to 30m Landsat data as mentioned in the suggested study.  Perhaps the issue of spatial transferability of models becomes a greater issue at fine scale mapping.  This issue has been noted in the revised manuscript.

Line 120 – Plant community classification (used here and throughout) can be conducted on many different levels of vegetation hierarchies (like the USNVC). Or you can ask other questions, like changes in productivity between years. Community type generally shifts when a system crosses a threshold from disturbance/stressors, through succession, etc. You need to define much better in this manuscript what you mean, and what you are looking at in regards to, for example, plant community classification vs. plant community species or productivity. A single Landsat image may work very well for some purposes, but for the more detailed questions like yours, multiple images may be required (although you did not

actually test accuracy differences between the number of images). You seem to be focused more on classification of functional groups rather than a community.

Plant community in the context of this study has been defined for clarity in the manuscript.

Line 121 – The three references here are specific studies (two over 10 years old), not reviews of plant classification studies. Maybe in the past it was more common to use a single time period, but seems now with increased computing power and the availability of the entire Landsat achieve, etc., it is very common for much more robust and multiple acquisition studies. Maybe phrase this more to acknowledge this evolution.

This has been changed in the manuscript.

Line 135 – Again, what is a plant community. Have you really found or are looking that the community changes? Or are you looking at species representation within a community, i.e. functional group dominance and shifts in this between years.

See response to reviewer one in the overview comments. A plant community is a collection of species within an area of a relatively uniform composition different from neighboring patches.  In this study, differences in neighboring patches are sometime evident in differences in dominant functional group (forb vs grass) or differences in photosynthetic pathways (C3 vs C4 grasses).  The aim of this study is not to measure community changes, as shifts in plant communities occur over longer time scales within perennial northern great plains plant communities.

Line 143 – NGP probably too broad for the implications of your study. Be more specific with the MLRA, or mixed grass prairie systems, etc. that you are testing. Intro – Lots of general and vague statements in here and limited citations supporting broad brush statements. I suggest going through the intro to make it more specific. For example, landscape, local, various etc. scales will mean different things to every reader. Define these or what they are for the studies you cite. Also make sure your statements are supported by citations or explained. For example, line 61 – 65 is not a summation or conclusion of the paragraph, so these new statements should be supported. Finally, you should mention this is part of a larger study looking at cattle use compared to prairie dog prevalence and impact to pastures, but the paragraph starting on line 130 had me confused between what this study was going to do, and what the larger study did.

Intro has been changed to include mixed grass prairie specifically, though much of the Northern Great Plains is comprised of mixed grass prairie.  Intro has been changed to specify the differences in scales between citations such as Landsat imagery versus high resolution imagery.

Additional text has been included to distinguish the objectives of this study and those of the larger study for clarity.

Line 174 – These sound more like plant functional groups than communities. Nothing wrong with mapping those, but the terminology issues are prevalent and I believe confuse your conclusions. Changes in representation are common between years, changes in community are a different boat.

See response above.

Line 199 – Why not compare them all together? You need to add rationale for why separating these out beforehand is appropriate. If you want to scale up your study, how will you separate out prairie dog towns at the "landscape" scale (watershed, county, etc.). As another option, I would find this much more compelling if the comparisons and RF models were tested to separate out all five groups. This would be a much more thorough test of RF.

The revised manuscript has been changed to test for differences between all five groups, not separated by on-town or off-town locations.

Line 217 – Do you think the wider spectral bands (compared to Landsat or UAV options) played into your results at all?

No

Line 227 – Again, this needs more justification then saying they are mutually exclusive. You either decided to map prairie dog towns separately from the rest of the study area (which you need to justify why) or you could test what the implications are of not having mapped towns in the first place (which also can vary between years).

See above

Line 239 – Why only 100, when the default is higher (which is used for the number of nodes)? You may be ok here, but in many cases, at this point the model error is just beginning to stabilize. You could examine the impact of the number of trees on your model by looking over a range of "number of tree" values.

The revised manuscript has increased the number of trees to 200.  Adding additional trees (default=500) was computationally prohibitive. For all models, the error was plotted to check for stabilization.  Most models had stabilized by 50 trees.  Previous research in remote sensing has demonstrated that the number of trees has little influence over classification results, and that ensembles of 70 trees are sufficient for classification.  Adding a large number of trees beyond error stabilization is unlikely to improve classification accuracy and will only add to computational time. See:

Du, P., A. Samat, B. Waske, S. Liu, Z. Li. Random forest and rotation forest for fully polarized SAR image classification using polarimetric and spatial features ISPRS J. Photogramm. Remote Sens., 105 (2015), pp. 38-53

Topouzelis K., A. Psyllos. Oil spill feature selection and classification using decision tree forest on SAR image data ISPRS J. Photogramm. Remote Sens., 68 (2012), pp. 135-143

Line 241 – Why just spectral bands as input into the models? You don't explicitly say your objective is to use just satellite imagery (and prior to the RF algorithm you used other data, e.g. to differentiated prairie dog towns and off site)

It is explicitly mentioned in the goals of the project to 'assess the utility of using a RF model with high resolution satellite imagery to classify plant communities'. Delineating prairie dog town boundaries with GIS would be akin to outlining the boundary of any other study area of interest.

Line 246 – How did you apply your models to produce predictions for prairie dog towns vs. off town locations? I think you run the predictions on two separate parts of the study area (be explicit).

The 'predict' function in Program R was used to apply the models. This has been added to the manuscript.

Line 262 – The way your "communities" were picked seem to almost guarantee this? You picked areas "dominated" by three (or two) very different functional groups. Is this overlap more than you expected, and what is the overlap? This very much may help explain the differences between years.

We would expect a large separation in ordination space based on how plant communities were selected. I think it is of value for plant classification studies to demonstrate that the plant communities one is classifying are actually distinct. The amount of overlap between plant communities may also factor into error rates or help explain differences between years. As mentioned prior, plant communities may be dominated by a specific functional groups, but other functional groups and species exist within these areas. Text has been added to the manuscript to reflect this.

Line 287 – Are the models unstable, or does this indicate the models are accurate within years, but species representation (as seen through your methods) changes between years in heterogenous areas?

Unsure what is meant by species representation. Given that these are perennial plant communities, it would be unlikely for major shifts in dominance to occur between successive years without a major disturbance event (i.e. fire).

Line 303 – These peaks seem like they very much may affect the production of warm vs. cool season grasses between the years as well.

Agreed, it could affect production to some extent.

Line 304 – Was there a temperature difference between years as well? These curves seem farther apart then I would expect just based on precip.

Temperature was similar between years. From the manuscript: 'Oesterheld et al. (2001) showed that annual above ground primary production of shortgrass communities is related to current as well as previous two years precipitation. The above average rainfall at the study site in 2015 could have added to the increase in average NDVI in 2016 when compared to 2015 through an increase in cumulative biomass or production at the site'.

Line 328 – Is there a transition zone at the edge of the prairie dog towns too?

Though there are transition zones at the edge of prairie dog towns, they tend to be much sharper boundaries and occur in off-town sites. This results in much more distinct boundaries and improves the ease of mapping colonies. The revised manuscript includes combining on and off town plant communities and highlights the distinct boundary between colonized areas and uncolonized areas.

Line 340 – Based on your discussion so far, what is a more accurate thematic map? Which year is the truth, if the heterogeneous transition zones may switch categories depending on which group dominates in a given year? How about comparing this map to the two yearly maps?

The map which includes both 2015 and 2016 data is likely the most accurate map, as demonstrated in the lower error rates. More information (spectral values across seasons and years) would produce a more accurate thematic map. As mentioned prior, switching in dominance, especially functional group dominance, between consecutive years is unlikely to occur in perennial mixed-grass prairie ecosystems without a major disturbance occurring.

Line 351 – Any limitations in the approach though? How about the lack of coefficients for your variables? I.e. good for prediction, not as good for understanding relationships

The goal of creating predictive models is to generate good predictions. The aim of this study was prediction, not inference.

Line 353 – Why not include the variable importance for the combined model?

This is included in the revised manuscript.

Line 562 – Break this out to be more specific on the changes per year (what was it in 2015 and what is it now in 2016) rather than lumping the switches between types that switch both ways in the two years. If there is a dominant pattern of switch that would be useful for your conclusions.

This is included in the revised manuscript.

Line 588 – How are the draws mapped? These are not one of your groups, need to talk about this in the methods. Technical Corrections Line 3 – Consider replacing stability, I think this could be confused with other definitions and is not quite what you mean

Draws were delineated visually via ArcGIS and imagery. This is included in the revised manuscript.

Line 32 – Replace highly with high Line 46 – Replace instability with disagreement Line 66 – Remove colonization and replace dog with dogs

This is included in the revised manuscript.

Line 86 – First time you use the acronym NDVI. Write out fully.

This is included in the revised manuscript.

Line 98 – Replace several with many (or similar idea)

This is included in the revised manuscript.

Line 101 – Replace proven with demonstrated

This is included in the revised manuscript.

Line 109 – There are a lot more RF packages and implementation options now, compared to 2013. Standard software like R, ERDAS Imagine, QGIS, and ArcGIS have RF, as well as more specialized options

like Ecognition (and even Google Earth Engine). I don't think you need this sentence, not relevant to the paper.

This has been deleted in the revised manuscript.

Line 173 – Need year you accessed the Mesonet data Line 181 – About how big are these (median, range, etc.)

This is included in the revised manuscript.

Line 192 – How were they randomly located?

This is included in the revised manuscript.

Line 236 – Did you consider other potential predictors that you could derive from these inputs?

We didn't include additional vegetation indices from the spectral data. Additional metrics could include elevation, ecological site, etc. though given the low training error from the models this would not likely change the predictions.

Line 239 – What is the default number of nodes. Define this.

This is included in the revised manuscript.

Line 256 – A table of the species for each of the five groups would really help. Would also help understand what "dominated" means for your training sites.

Species are listed in the site description

Line 267 – Mishra and Crews should be outside parentheses

This is included in the revised manuscript.

Line 310 – What was the 2014 precip then? Dry?

Wet

Line 355 – For the town or off-site model?

The study had been changed to include both off-town and on-town communities into the model.

**Response to Reviewer 2**

The authors use the unique plant community signature of Prairie Dog colonies to challenge RF methods, but the novelty of this approach is never articulated. Explain early on, with references, why temporal and spatial characteristics of prairie dog influence on vegetation makes it an interesting challenge for remote sensing and the combined ecological/rangeland management/remote sensing triumvirate of the manuscript will be clearer to the reader. The Introduction needs to be restructured and I recommend the Results and Discussion be entirely re-written, it was extremely difficult to follow and all of the cool aspects of this interesting study were either buried or not mentioned at all. After rather major revisions I can see how this paper could be acceptable for publication. It is technically sound for the most part but needs major changes.

The introduction has been restructured in the revised manuscript. The results and discussion have been changed to improve clarity in reading the manuscript. This includes combing all plant communities into the model instead of separating them into on-town and off-town. Sub headers have been included in the results and discussion as well for clarity, and a greater focus of the manuscript has been placed on remote sensing prairie dog colonies.

Minor comments: The ecological justification for investigating Prairie Dog towns was somewhat lacking in the abstract. Is this study fundamentally about identifying colonies from remote platforms or using prairie dog colonies as an interesting opportunity to advance statistical techniques in remote sensing?

The study was part of a larger study focusing on livestock production and diet selection within pastures occupied by prairie dog towns. Additional analysis has been included to highlight the ability of remote sensing to map prairie dog colonies.

The statement on line 43 is somewhat fuzzy. The cautious note at the end of the abstract is forthcoming.

The sentence has been changed in the manuscript.

The transition from line 65 to 66 is a bit harsh. The narrative 'funnels' from remote sensing in general to prairie dog colonies in particular far too rapidly. As a consequence, the reader is left wondering if the central theme is prairie dog colony identification or remote sensing techniques or rangeland and cattle management (or all of the above, and if so how do they fit together).

The introduction has been restructured to add clarity.

The paragraph beginning line 79 is 'listy' and reads like a few random manuscripts that the authors read. How do these fit together to advance the overall objective of the study? I recommend restructuring the Introduction. 'Writing Science' by Schimel is a good text for describing logical flow in scientific manuscripts.

These manuscripts fit into the overall objective of the study by demonstrating mapping of various plant communities using spectrally derived data from satellite imagery, which is the overall goal of this study.

From the paragraph on line 101 it appears that the objective isn't to compare RF against different techniques, which is fine. But the opportunity to use the subtle (or not so subtle) vegetation changes induced by prairie dog colonies to challenge RF methods isn't brought to the forefront. This is a missed opportunity in my opinion. Note also in line 146 that a goal could also be to investigate prairie dog and plant ecology: you don't always have to bring it back to cattle foraging. The Utah and Mexican Prairie Dogs are endangered after all.

Numerous studies have investigated prairie dog impacts of plant ecology.  Prairie dogs can have a large impact plant species composition, and older core areas often become characterized by annual forbs species and low production, which can directly impact livestock production.  Additional analysis has been included in the revised manuscript to include mapping prairie dog colonies via remote sensing.

156: The Ecological Sites notion was new to me and the descriptions sound like soil types. Are these a USDA thing?

Ecological sites are used heavily by USDA agencies.  They are a distinctive type of land with specific soil and physical characteristics produce unique vegetation.

162: I'm confused, I always thought that Kentucky bluegrass was Poa pratensis.

This has been corrected in the revised manuscript.

173: the temperature and precip measurements are great but please specify the mesonet used (South Dakota).

This has been corrected in the revised manuscript.

174: using common abbreviations like 'pdf' or common words like 'snow', 'cool', and 'warm' will lead to confusion. Sites are either on towns or off, so using PD with subscript f or g, then O (or similar, even 'NPD' as used on line 201 without previous description) with subscripted snowberry, c3, and c4 would help me at least. There is a lot to digest here and making things easier for the reader can go a long way.

Acronyms have been changed to improve clarity for the reader.

I'm not entirely sure why an ordination, MRPP, NMS, Bray-Curtis, etc. was used for pre-defined vegetation types. Weren't they already selected to be different from each other? Is the point of this analysis to guarantee that the five vegetation types are in fact different from each other (e.g. line 256)? In this case of course it's fine to do so.

We would expect a large separation in ordination space based on how plant communities were selected. I think it is of value for plant classification studies to demonstrate that the plant communities one is

classifying are actually distinct.  The amount of overlap between plant communities may also factor into error rates or help explain differences between years.

NDVI probably doesn't need to be defined on 231 although a note about any differences in the spectral resolution of the red and NIR among Pleiades and other common satellites may be interesting for the Discussion.

NDVI definition has been removed.

276 is probably a methods point and 278 may even be an Introduction point. Literature as a whole needs to be woven into the narrative. In general, any time a sentence starts with the author of a paper, the sentence needs to be changed. Doing this makes the author(s) the subject(s) of the sentence. The topic at hand should be the topic of the sentence. Please start a sentence with authors only when those authors are the subject of the sentence, which can happen.

Listing an author at the beginning of a sentence is a common convention in ecological literature.

The paragraph beginning 265 could benefit from a few more quantitative values rather than qualitative ones like 'high degree' and 'lower'.

Tables are referenced giving specific values. Numbers are generally either referred to in tables or the text but not both.

296: I disagree somewhat. Different species will be more prominent during different times of the year (e.g. cool vs warm season grasses).

I agree somewhat that different species will be more prominent during different times of the year, this is especially true of forb species that flush early in the season.  However with perennial cool or warm season grasses, they still occupy the same physical space on the landscape, just differ phonologically.

The manuscript would probably benefit from separating the results and discussion to show first what happened then explain it. The discussion never comes back to prairie dogs.

Sub headers have been included in the results/discussion to add clarity.  Additional analysis has been included at the end to bring the discussion back to prairie dogs.

Please make font sizes larger in the figures. They are often hard to read.

This has been corrected in the revised manuscript.

From Fig. 5 and 6 it appears that prairie dog colonies, at least in this area of SD, can be identified with a relatively large degree of accuracy. This needs to be made more prominent in the discussion.

This has been addressed in the revised manuscript.

---

## Referee Report (RR1)

Revision Review for bg-2019-194:
Comparing Stability in Random Forest Models to Map Northern Great Plains Plant Communities in Pastures Occupied by Prairie Dogs Using Pleiades Imagery

Overview

Thanks for the revised manuscript and for your care and consideration of peer review comments. I find this revised manuscript to be much clearer and that the analyses now support the objectives and purpose laid out in the introduction. These changes have notably improved the manuscript and have removed confusion in the interpretation of findings. I have mostly clarification and technical comments that I feel will aid readers in understanding this work.

Specific Comments

Line 101 – If I understand the conclusions from Juel et al. 2015, then one logical extension would be that we also need to consider having spatially relevant training data (i.e. to address your issue that models may not transfer in space and time). Consider adding some additional possible solutions and implications of classification schemes (e.g. cover amounts of functional groups vs. community type)

Line 133 – I find the connection between "signatures on imagery" and plant community response to the timing and progression factors underdeveloped. Add a sentence or two expanding what specifically will change within your communities (with relatively uniform composition) within and between years. Maybe a specific example would help here too.

Line 288 – Any spatial consistency on where these are? I.e. do they represent edges of the community where precip changes may lead to this finding? Would support next few sentences.

Line 317 – Talk about what this this means in terms of changes in or between your community types

Line 329 – Need some discussion about how the selection of your community types leads to some heterogeneity within types, but this is a needed tradeoff (to lead into next paragraph)

Line 393 – You have assessed the accuracy based on your 2016 data. So additional years helped you accurately predict your training sites from 2016 (relatively homogeneous areas). Be specific about what accuracy you have measured, which really is model performance here.

Line 398 – Do you also mean here that the selection of community types to map is an important consideration. I know you did not explore this specifically but seems to be an important theme in your discussion and results. Add some discussion and concluding statements about this aspect.

Line 610 – Here and throughout the Tables and Figures please check and revise for acronym consistency. You switch between On-PDG and On-Grass, and On-PDF and On-Forb, within and between figures and tables

Technical Comments

Line 93 – Need parentheses around 2015

Line 99 – Parentheses around 2018(check rest of document for formatting of refs too)

Line 111 – Do you mean prairie systems worldwide or specifically mixed grass prairies of the U.S. Northern Great Plains? I think you need to be specific here of the geographic region this paragraph addresses.

Line 125 – If you have a ref to send readers to about the larger study, please add.

Line 171 – Add the specific station used and check citation info (I found/used ref below).

South Dakota Mesonet, South Dakota State University. (2019). *South Dakota Mesonet Database* [database].

Line 179 – Last sentence probably not needed. Also consider moving sentences (lines 225-227) about removing these areas and mapping prairie dog colonies up to this spot for reader clarity.

Line 270 – For consideration, is "error" the best term here? For the message in your manuscript maybe use "instability?" You have the common problem of heterogeneity in your pixels/plots which makes it hard to classify to a specific type and your analysis shows that the year used can switch these mixed pixels between classes (the stability issue you are covering).

Line 298 – Nice discussion in this paragraph

Line 301 – Need reference

Line 400 – Clarify that this is transition "zones" between communities (and not through time)

Line 621 – The locations of the on plot labels were confusing to me at first. Consider making these the same color as the community points and in the figure legend discuss what the +'s in the plot represent (this may also help folks identify the labels go with these centers)

Line 667 – Check acronyms for consistency (see comment on 610)

Line 683 – Check plot labels (see comment on 610)

---

## Author Response (AR2)

Response to Reviewer 1

Revision Review for bg-2019-194:

Comparing Stability in Random Forest Models to Map Northern Great Plains Plant Communities in Pastures Occupied by Prairie Dogs Using Pleiades Imagery

Overview

Thanks for the revised manuscript and for your care and consideration of peer review comments. I find this revised manuscript to be much clearer and that the analyses now support the objectives and purpose laid out in the introduction. These changes have notably improved the manuscript and have removed confusion in the interpretation of findings. I have mostly clarification and technical comments that I feel will aid readers in understanding this work.

Specific Comments

Line 101 – If I understand the conclusions from Juel et al. 2015, then one logical extension would be that we also need to consider having spatially relevant training data (i.e. to address your issue that models may not transfer in space and time). Consider adding some additional possible solutions and implications of classification schemes (e.g. cover amounts of functional groups vs. community type)

Some additional text has been included on Line 95 to discuss this.

Line 133 – I find the connection between "signatures on imagery" and plant community response to the timing and progression factors underdeveloped. Add a sentence or two expanding what specifically will change within your communities (with relatively uniform composition) within and between years. Maybe a specific example would help here too.

This sentence has been removed.  The connection between signatures on the imagery and plant community response is discussed in greater detail in the results and discussion section.  See paragraph beginning on page 304.

Line 288 – Any spatial consistency on where these are? I.e. do they represent edges of the community where precip changes may lead to this finding? Would support next few sentences.

See additional text:

These are likely occurring along transition zones between prairie dog colony edge.

Line 317 – Talk about what this this means in terms of changes in or between your community types

See additional text:

Increased cumulative biomass in 2016 may cause higher NDVI values for example in On-PDG plant communities resulting in classification shifts to Off-Cool; similarly, greater NDVI values in Off-cool in 2016 may result in some of those pixels being classified as Off-Snow.

Line 329 – Need some discussion about how the selection of your community types leads to some heterogeneity within types, but this is a needed tradeoff (to lead into next paragraph)

Paragraph has been re-structured, and selection of plant communities and changes within types brought back to prairie dog influence of vegetation.

Line 393 – You have assessed the accuracy based on your 2016 data. So additional years helped you accurately predict your training sites from 2016 (relatively homogeneous areas). Be specific about what accuracy you have measured, which really is model performance here.

Point noted, this has been changed to model performance.

Line 398 – Do you also mean here that the selection of community types to map is an important consideration. I know you did not explore this specifically but seems to be an important theme in your discussion and results. Add some discussion and concluding statements about this aspect.

Additional text added:

…recognizing that plant communities rarely exist in discrete communities is important when selecting community types to map.  Combining plant community ordination results with remote sensing results can aid in understanding sources of model error and limitations of classification algorithms.

Line 610 – Here and throughout the Tables and Figures please check and revise for acronym consistency. You switch between On-PDG and On-Grass, and On-PDF and On-Forb, within and between figures and tables

Corrected

Technical Comments

Line 93 – Need parentheses around 2015

Corrected

Line 99 – Parentheses around 2018(check rest of document for formatting of refs too)

Corrected

Line 111 – Do you mean prairie systems worldwide or specifically mixed grass prairies of the U.S. Northern Great Plains? I think you need to be specific here of the geographic region this paragraph addresses.

Changed to norther great plains mixed grass prairie

Line 125 – If you have a ref to send readers to about the larger study, please add.

added

Line 171 – Add the specific station used and check citation info (I found/used ref below). South Dakota Mesonet, South Dakota State University. (2019). South Dakota Mesonet Database [database].

added

Line 179 – Last sentence probably not needed. Also consider moving sentences (lines 225-227) about removing these areas and mapping prairie dog colonies up to this spot for reader clarity.

Sentence removed and additional sentence moved up in text.

Line 270 – For consideration, is "error" the best term here? For the message in your manuscript maybe use "instability?" You have the common problem of heterogeneity in your pixels/plots which makes it hard to classify to a specific type and your analysis shows that the year used can switch these mixed pixels between classes (the stability issue you are covering).

Point noted, changed to instability

Line 298 – Nice discussion in this paragraph

Line 301 – Need reference added

Line 400 – Clarify that this is transition "zones" between communities (and not through time)

added

Line 621 – The locations of the on plot labels were confusing to me at first. Consider making these the same color as the community points and in the figure legend discuss what the +'s in the plot represent (this may also help folks identify the labels go with these centers)

Added in the legend is what +'s mean in the plot.

Line 667 – Check acronyms for consistency (see comment on 610)

Corrected

Line 683 – Check plot labels (see comment on 610)

Corrected.

Response to Reviewer 2

A number of improvements were made but I still had a difficult time reading the manuscript. The response 'Listing an author at the beginning of a sentence is a common convention in ecological literature' assumes that the ecological literature is written well. It largely is not. For this and other reasons the paragraph beginning on line 64 should be cut in its entirety and I revert to Josh Schimel's recommendation that every sentence that begins with an author needs to be rewritten if the authors are not the subject of the sentence. Doing so will make the authors realize that the structure of the text needs to change to have a simpler logical flow that makes it more apparent why prairie dog colonies make for interesting remote sensing challenges. Much of the introduction reads like a grab-bag of random papers and I don't feel that my suggestion to improve it was taken seriously. The Results and Discussion are better but could use further improvement. These are important points because it would be nice if this interesting study was more readable.

The introduction has been restructured per recommendations, this includes bringing the significance on why mapping prairie dog plant communities are an important task ecologically to the forefront of the article. Additionally text in the introduction has been added to highlight why vegetation changes associated with increased herbivory from prairie dogs might pose challenges to remote sensing. In addition sentences in the introduction, results and discussion have been changed unless the author is the subject.  Significant portions of the results and discussion has been re-structured for clarity. Additional discussion has included impacts prairie dog herbivory have on plant communities and our ability to detect these with satellite imagery.

The abstract would benefit from a brief discussion of why prairie dog colonies are important keeping in mind the international and/or non-ecological readership who would benefit from an explanation. In brief, the manuscript is technically sound but won't reach the intended audience unless the reader can see more clearly how interesting it is.

Additional discussion has been included in the abstract as well as the introduction about why prairie dogs are important to reach a broader audience.

Minor point?
152: How did cattle impact vegetation?

Cattle can impact vegetation, but this is dependent at the intensity of grazing.  At 50% utilization, livestock will have a minimal impact on plant communities.  Within the larger study no difference in plant communities were detected between off-town plots where cattle were excluded and off-town plots where cattle were allowed to graze.  Differences largely align along an on- town and off-town gradient.

[revised manuscript text omitted]